# Why Are *Linear* RNNs More Parallelizable?

William Merrill [1]   Hongjian Jiang [2]   Yanhong Li [1]   Anthony Lin [2 3]   Ashish Sabharwal [1]

## Abstract

The community is increasingly exploring linear RNNs (LRNNs) as language models, motivated by their expressive power and parallelizability. While prior work establishes the expressivity benefits of LRNNs over transformers, it is unclear what makes LRNNs—but not traditional, *nonlinear* RNNs—as easy to parallelize in practice as transformers. We answer this question by providing a tight connection between types of RNNs and standard complexity classes. We show that LRNNs can be viewed as log-depth (bounded fan-in) arithmetic circuits, which represents only a slight depth overhead relative to log-depth boolean circuits that transformers admit. Furthermore, we show that nonlinear RNNs can solve L-complete problems (and even P-complete ones, under polynomial precision), revealing a fundamental barrier to parallelizing them as efficiently as transformers. Our theory also identifies fine-grained expressivity differences between recent popular LRNN variants: permutation-diagonal LRNNs are $NC^1$-complete whereas diagonal-plus-low-rank LRNNs are more expressive ($PNC^1$-complete). We provide further insight by associating each type of RNN with a corresponding automata-theoretic model that it can simulate. Together, our results reveal fundamental tradeoffs between nonlinear RNNs and different variants of LRNNs, providing a foundation for designing LLM architectures that achieve an optimal balance between expressivity and parallelism.

## 1. Introduction

Parallelism and expressive power are both desirable properties in an LLM architecture that are, unfortunately, at odds

[1]Allen Institute for AI [2]Rheinland-Pfälzische Technische Universität [3]Max-Planck Institute for Software Systems. Correspondence to: William Merrill <willm@allenai.org>.

*Proceedings of the 43rd International Conference on Machine Learning*, Seoul, South Korea. PMLR 306, 2026. Copyright 2026 by the author(s).

(Merrill & Sabharwal, 2023). RNN architecture design reflects this tradeoff: while older RNNs were nonlinear and highly sequential (Elman, 1990; Hochreiter & Schmidhuber, 1997), more recent RNN architectures prefer a *linear* state update (Bradbury et al., 2017; Katharopoulos et al., 2020; Gu & Dao, 2024, inter alia) to enable parallel processing of long sequences (Blelloch, 1990). Linear RNNs (LRNNs) have been shown to be surprisingly theoretically expressive relative to transformers (Merrill et al., 2024; Grazzi et al., 2025) and practically performant (Beck et al., 2024; Yang et al., 2025; Peng et al., 2025), but it remains unclear how LRNNs and nonlinear RNNs compare in expressive power, or, conversely, whether fundamental barriers prevent parallelizing nonlinear RNNs to the same degree as LRNNs.

In this work, we theoretically compare the parallelizability and expressive power of nonlinear RNNs and LRNNs through the lens of computational complexity theory. A core result is a conditional separation in expressive power between the two classes: whereas LRNNs can only solve tasks that are in the circuit complexity class $PNC^1$ (Theorem 3), nonlinear RNNs can express P-complete problems with polynomial precision (Corollary 2) or L-complete problems when restricted to log precision (Theorem 2). In either regime, this suggests nonlinear RNNs can represent substantially less parallelizable computation that's beyond the capabilities of LRNNs, assuming $PNC^1 \neq P$ with poly precision, or $PNC^1 \neq L$ with log precision.

Conversely, these expressivity results have implications for the parallelizability of different types of RNNs. It follows that LRNNs—regardless of precision—can be simulated by NC circuits of depth $O(\log n \log^* n)$, i.e., with a negligible depth overhead of $O(\log^* n)$ beyond the wall-clock time we would expect for simulating a transformer. In contrast, our results imply that nonlinear RNNs with log precision likely require $\Omega(\log^2 n)$ depth, constituting a $O(\log n)$ overhead relative to a transformer. Moreover, with polynomial precision, nonlinear RNNs require more than polylog depth to simulate with bounded fan-in circuits, assuming $NC \neq P$.

Beyond comparing nonlinear RNNs and LRNNs, we also compare the fine-grained expressivity of different LRNN variants within $PNC^1$. In particular, we consider differences between several different parameterizations that can express $NC^1$-complete problems, showing that diagonal-plus-low-

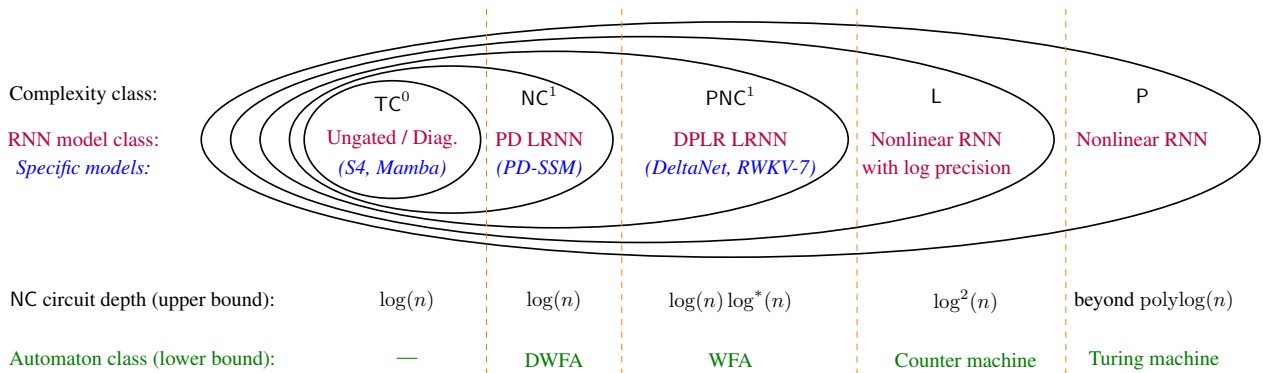

*Figure 1.* Main results, summarized as a hierarchy of increasingly expressive RNN classes with popular *models* within each class. Each RNN shown is "tight" for the respective complexity class C ($\mathsf{PNC}^1$, L, etc.) in the sense that both falls in C and can solve a C-complete problem. LRNNs are in $\mathsf{PNC}^1$, implying they can be nearly as efficiently parallelized as transformers, incurring only a small $O(\log^*(n))$ depth overhead in terms of bounded fan-in boolean circuits. Nonlinear RNNs, in contrast, can solve L-complete and even P-complete problems, but this comes at the cost of being less parallelizable, requiring notably deeper circuits. Bottom row lists the classes of automata that each RNN model class can simulate, where WFA stands for weighted finite automaton and DWFA stands for deterministic WFA.

rank (DPLR) LRNNs like DeltaNet (Yang et al., 2024; Grazzi et al., 2025) and RWKV-7 (Peng et al., 2025) can express $\mathsf{PNC}^1$-complete problems, whereas permutation-diagonal (PD) LRNNs (Terzic et al., 2025) are in $\mathsf{NC}^1$. These results, combined with our other results about nonlinear RNNs, yield a comprehensive hierarchy of the expressivity landscape for RNNs that is summarized in Figure 1.

Beyond our complexity characterization, we also associate each RNN and LRNN class with a corresponding automata-theoretic model from prior work (Peng et al., 2018; Weiss et al., 2018; Merrill et al., 2020) that this class can simulate.

Finally, we find empirically that our results [1] about RNNs' expressivity predict their behavior when trained on synthetic tasks. On an L-complete graph connectivity task, only nonlinear RNNs achieve good performance, as theory predicts. In addition, both nonlinear RNNs and DPLR LRNNs learn iterated matrix multiplication problem ($\mathsf{PNC}^1$-complete), but transformers and Mamba cannot, in line with theory. These tasks provide potential synthetic benchmarks for LRNN architecture development beyond the traditional state tracking and recall tasks that are currently popular (e.g., Arora et al., 2024; Beck et al., 2024; Peng et al., 2025).

Overall, our theoretical results reveal the fundamental trade-offs in expressivity and parallelism between linear and non-linear RNNs. Moreover, we show fine-grained differences in expressivity between LRNNs. Taken together, we hope our results can help the community continue to refine LRNN architectures to push the frontier between expressivity and parallelism as much as possible up to fundamental barriers.

---

[1] Code link: https://arg-git.informatik.uni-kl.de/pub/LinearRNN

## 2. Preliminaries

We first introduce the datatypes we use to model bounded-precision arithmetic, as well as various LRNN architectures. We then turn to introducing computational models from circuit complexity and and formal language theory that will be relevant in our analysis of LRNNs.

### 2.1. Datatypes

Analyzing LLM architectures via circuit complexity requires making assumptions about the underlying datatype. Prior work has considered various options while trying to ensure nice numerical properties like associativity and commutativity (Merrill et al., 2024). In this work, guided by the theory of weighted finite automata, we make these desiderata more explicit by asserting that the numeric datatype should be a semiring, denoted $\langle \mathbb{K}, \oplus, \otimes \rangle$. That is, addition and multiplication should be associative with identities, addition should commute, and multiplication should distribute over addition This makes the output of LRNNs more cleanly defined in an implementation-agnostic way, and also makes WFA and matrix multiplication computations well-defined for these datatypes (which will be important).

In particular, we will work with the standard *rational* semiring used in prior analysis, (i.e., polynomial-precision rational numbers; Chiang, 2025), though other semirings are possible in principle.

**Definition 1.** Rational numbers, $\mathbb{Q}$, form the **rational semiring** with the standard arithmetic semantics for $\oplus$ and $\otimes$.

We assume $\mathbb{Q}$ admits an **order** $>$ in the standard way. Additionally, we equip $\mathbb{Q}$ with division in order to capture layer normalization Finally, we define non-linear functions $\sqrt{\cdot}$ and $\exp$ via their Taylor series approximation in terms of

$\oplus$ and $\otimes$ to $O(\text{poly}(n))$ terms (cf. Chiang, 2025). Each of these operations can be given poly- or log-precision variants, which return an $(n^c)$- or $(c \log n)$-bit approximation using this technique, respectively (see Definition 10 ).

A natural notion for describing "simple" arithmetic functions will be the following:

**Definition 2** (Arithmetic Closed Form). $f : \mathbb{K}^k \to \mathbb{K}$ has an *arithmetic closed form* if it can be written as a finite composition of $\oplus, \otimes, \max, \exp$, and $\sqrt{\cdot}$ as defined for $\mathbb{K}$.

The following property about closed forms will be useful:

**Lemma 1** (Chiang, 2025). *If $f : \mathbb{Q}^k \to \mathbb{Q}$ has an arithmetic closed form, then $f$ can be computed in* FO-*uniform* $\mathsf{TC}^0$.

## 2.2. Nonlinear RNNs

An RNN over a datatype $\mathbb{K}$ (generally $\mathbb{Q}$) is a neural network component that takes as input a sequence of vectors $x_1, \dots, x_n \in \mathbb{K}^d$ and outputs a new sequence $y_1, \dots, y_n \in \mathbb{K}^d$ that sequentially mixes them. Classical RNNs (Elman, 1990; Hochreiter & Schmidhuber, 1997) were defined by updating a recurrent state vector via the application of some (generally nonlinear) function:

**Definition 3** (Nonlinear RNN). Given $\mathbf{x}_1, \dots, \mathbf{x}_n \in \mathbb{K}^d$, an RNN computes a a sequence of recurrent states $\mathbf{h}_1, \dots, \mathbf{h}_n \in \mathbb{K}^d$ via $\mathbf{h}_t = f(\mathbf{h}_{t-1}, \mathbf{x}_t)$, where $f$ is a closed-form arithmetic function (cf. Definition 2).

We consider two nonlinear RNN parameterizations. First, *ReLU RNNs* are nonlinear RNNs with $f(\mathbf{h}_{t-1}, \mathbf{x}_t) = \max\{\mathbf{A}\mathbf{h}_{t-1} + \mathbf{B}\mathbf{x}_t, 0\}$, where $\mathbf{A}$ and $\mathbf{B}$ are matrices $\mathbb{K}^{d \times d}$. More generally, we will sometimes consider the more general *MLP RNN* where $f$ can be a multi-layer feedforward network using ReLU activations.

## 2.3. Linear RNNs

Going beyond classical RNNs, there has recently been interest in RNNs with *linear* state updates due to the advantages for parallelizing on the sequence dimension. These linear RNNs (LRNNs) can be defined as follows:

**Definition 4** (LRNN). Given $\mathbf{x}_1, \dots, \mathbf{x}_n \in \mathbb{K}^d$, an LRNN head computes a sequence of matrix valued states $\mathbf{S}_1, \dots, \mathbf{S}_n \in \mathbb{K}^{d \times d}$ via

$$\mathbf{S}_t = A_t(\mathbf{x}_t)\mathbf{S}_{t-1} + b_t(\mathbf{x}_t),$$

where $A_t(\mathbf{x}_t) \in \mathbb{K}^{d \times d}$ and $b_t(\mathbf{x}_t) \in \mathbb{K}^d$ have arithmetic closed forms independent of $t$.

Definition 4 leaves open the specific parameterization of $A_t$ and $b_t$ as a function of $\mathbf{x}_t$. The choice of parameterization is crucial for theoretical expressivity (Merrill et al., 2024) as well practical concerns like memory. For instance,

constrained parameterizations (time-invariant or diagonal) were explored in early LRNNs, but these were shown to limit expressivity to $\mathsf{TC}^0$ (Merrill et al., 2024). Since then, newer LRNNs have adopted richer parameterizations that are non-diagonal but still efficient. Two broad classes of "slightly non-diagonal" parameterizations are diagonal-plus-low-rank (DPLR) and permutation-diagonal (PD) LRNNs.[2]

**Definition 5** (DPLR). An LRNN is *DPLR* if it satisfies $\mathbf{A}_t = \mathbf{D}_t - \mathbf{k}_t^\top \mathbf{v}_t$, where $\mathbf{D}_t$ is diagonal, $\mathbf{k}_t, \mathbf{v}_t$ are vectors, and all have an arithmetic closed form in terms of $\mathbf{x}_t$.

In particular, we consider specific DPLR variants:

- **RWKV-7** (Peng et al., 2025):

  $$\mathbf{A}_t = \text{diag}(\mathbf{w}_t) - \lambda_t \kappa_t (\mathbf{a}_t \odot \kappa_t)^\top \quad \mathbf{b}_t = \mathbf{v}_t \tilde{\mathbf{k}}_t^\top,$$

  for per-token vectors $\mathbf{w}_t, \mathbf{a}_t, \kappa_t, \mathbf{v}_t, \lambda_t, \tilde{\mathbf{k}}_t \in \mathbb{K}^d$ are produced from $\mathbf{x}_t$ via linear projection.

- **DeltaNet** (Yang et al., 2024):

  $$\mathbf{A}_t = \mathbf{I} - \beta_t \mathbf{k}_t^\top \mathbf{k}_t \quad\quad \mathbf{b}_t = \mathbf{v}_t \mathbf{k}_t^\top,$$

  where $\mathbf{k}_t, \mathbf{v}_t, \beta_t$ depend on $\mathbf{x}_t$ via projection.

A key difference between RWKV-7 and DeltaNet is that the state update is symmetric in DeltaNet. Related to this, Grazzi et al. (2025) show how, for fixed-precision DeltaNet, the range of $\beta_t$ is crucial for expressivity: restricting $\beta_t \in (0, 1)$ prevents recognizes parity, but $\beta_t \in (0, 2)$ allows it. Both architectures have recently been trained at large scale (Peng et al., 2025; Kimi Team, 2025).

**Definition 6** (PD; Terzic et al., 2025). An LRNN is *PD* if it satisfies $\mathbf{A}_t = \mathbf{P}_t \mathbf{D}_t$ for a column one-hot 0-1 matrix $\mathbf{P}_t$ and a diagonal matrix $\mathbf{D}_t$, both with an arithmetic closed form in terms of $\mathbf{x}_t$.

$\mathbf{P}_t$ represents a function $\pi : \{1, 2, \dots, d\} \to \{1, 2, \dots, d\}$. Despite the name, $\mathbf{P}_t$ need not represent a permutation matrix, though it will be a permutation when $\mathbf{P}_t$ is full rank.

## 2.4. Multilayer RNNs

In line with current practice (Yang et al., 2025), we will imagine that these RNN components (Sections 2.2 and 2.3, respectively) are used as "heads" within a larger architecture resembling the transformer. A head is computed by passing $\mathbf{x}_1, \dots, \mathbf{x}_n$ to each head; the specific parameterizations mentioned above will compute quantities resembling keys and values in the transformer. After this, the head output $\mathbf{y}_t$ is determined as to be $\mathbf{h}_t$ in a nonlinear RNN and $\mathbf{q}_t^\top \mathbf{S}_{t-1}$ in an LRNN, where $\mathbf{q}_t = \mathbf{Q}\mathbf{x}_t$ is analogous to a query.

The overall architecture of a multilayer RNN then works as follows. The first layer maps each token to an embedding.

---

[2]Henceforth, for brevity, we will write $\mathbf{A}_t, \mathbf{k}_t$, etc. to mean $A_t(\mathbf{x}_t), k_t(\mathbf{x}_t)$, etc., leaving the dependence on $\mathbf{x}_t$ implicit.

This is followed by alternating LRNN sublayers, which mix information across tokens, and feedforward sublayers, which apply local computation at each token. More formally, these sublayers have the form:

**Definition 7** (Multihead RNN Sublayer)**.** Given $\mathbf{x}_1, \ldots, \mathbf{x}_n \in \mathbb{K}^d$, we first apply layer-norm, compute heads $\mathbf{y}_1, \ldots, \mathbf{y}_h$ in parallel, and then aggregate the outputs. Formally, the sublayer computes the mapping $\mathbf{x}_t \mapsto \mathbf{x}_t + \mathbf{O} \cdot \langle \mathbf{y}_1, \ldots, \mathbf{y}_h \rangle$.

**Definition 8** (Feedforward Sublayer)**.** Given $\mathbf{x}_1, \ldots, \mathbf{x}_n \in \mathbb{K}^d$, we first apply layer-norm and then apply a standard two-layer ReLU network. Letting $\tilde{\mathbf{x}} = \mathrm{lnorm}(\mathbf{x})$, the feedforward sublayer computes $\mathbf{x}_t \mapsto \mathbf{x}_t + \mathbf{W} \cdot \mathrm{ReLU}(\mathbf{U}\tilde{\mathbf{x}}_t)$.

A full RNN network consists of serial composition of multihead RNN and feedforward sublayers.

**Definition 9** (Language Recognition)**.** Given a datatype $\mathbb{K}$, we define the language recognized by an RNN as follows. Let $\$ \in \Sigma$ be a beginning-of-sequence symbol. For every string $w \in \Sigma^n$, we pass $\$w$ through the RNN and apply a linear transformation $\mathbf{o}$ to the final layer output $\mathbf{y}_n^\ell$ at the last token and accept if and only if $\mathbf{o}^\top \mathbf{y}_n^\ell > 0$.

**Precision.** Let $f : \Sigma^* \to \mathbb{K}^d$ be a representation in a neural sequence model (e.g., a transformer or RNN). We can define the precision of this representation as follows:

**Definition 10.** Let encode be the function that serializes each value in a vector $\mathbf{x} \in \mathbb{K}^d$ in binary. The precision of $\mathbf{h} : \Sigma^* \to \mathbb{K}^d$ is $p_\mathbf{h}(n) = \max_{w \in \Sigma^n} |\mathrm{encode}(\mathbf{h}(w))|$.

We define the precision of an RNN as the maximum precision over all components in its computation graph. By definition, all RNN over $\mathbb{Q}$ have a poly-size computation graph and hence at most *poly-precision*, i.e., $O(n^c)$ for some $c$. Some RNNs attain *log precision*, i.e., satisfy $O(\log n)$. When talking about log-precision networks, we will assume that log-precision variants of $\exp$ and $\sqrt{\cdot}$ are used in the computation graph, so that extra precision is not added to intermediate values needlessly.

## 2.5. Circuit Complexity

We will leverage circuit complexity (Vollmer, 1999) to formalize the tradeoff between parallelizability and expressivity of RNNs. We first recall standard circuit complexity classes. A *circuit* is a directed acyclic graph where leaf nodes represent input variables and internal nodes represent logic gates (e.g., AND, OR, NOT) applied to their children; a circuit $C_n$ define a boolean function $\{0,1\}^n \to \{0,1\}$, or, more generally, a function $\Sigma^n \to \{0,1\}$ for some finite vocabulary $\Sigma$, in the standard way. A *circuit family* $\mathcal{C} = \{C_n\}_{n=0}^\infty$ is then a collection of circuits indexed by input length that defines a formal language $L \subseteq \Sigma^*$ where $w \in L$ if and only if $C_{|w|}(w) = 1$. $\mathsf{NC}^d$ is the class of

languages recognized by bounded-fan-in boolean circuits of polynomial size and depth $O(\log^d n)$. Write $\mathsf{NC}$ for $\bigcup_{d \geq 0} \mathsf{NC}^d$. $\mathsf{AC}^d$ is the extension of $\mathsf{NC}^d$ that allows unbounded fan in, and $\mathsf{TC}^d$ extends $\mathsf{AC}^d$ by allowing majority (MAJ) gates in addition to AND, OR, and NOT.

In general, circuit families are a non-uniform model of computation: since there is a different circuit for each input length, there is no finite description of the overall computation. To rectify this issue, we can define *uniform variants* of circuit classes where the structure of $C_n$ is determined as as a function of $n$ via some kind of computable procedure. There are different natural variants of circuit classes with different levels of uniformity; unless otherwise specified, we work with first-order (FO) uniformity. See Strobl et al. (2024) for more background on these classes.

We will focus on the *bounded fan-in* classes, which are more aligned with real hardware. The expressivity of transformers (Hao et al., 2022; Merrill & Sabharwal, 2023) and simple LRNNs like Mamba (Merrill et al., 2024) is upper bounded by $\mathsf{TC}^0$, which is an unbounded fan-in class but known to be contained in the log-depth bounded fan-in class $\mathsf{NC}^1$. This latter class is **efficiently parallelizable** in the sense that problems in it can be solved by an appropriate hardware implementation in time proportional to the depth of the corresponding circuits, i.e., in logarithmic wall-clock time. With today's LLMs often trained for context lengths of 64K to 1M tokens, log depth is proportional to 16 to 20 whereas the log-squared depth of $\mathsf{NC}^2$ circuits is proportional to 256 to 400. Thus, log-squared depth increases sequential runtime by a factor of 16 to 20 relative to log depth, rendering the resulting computation significantly less parallelizable. We thus treat log-depth as the *boundary of efficient parallelization*.

A more liberal notion of efficient parallelizability allows for *arithmetic circuits* instead of boolean ones. For our purpose, an arithmetic circuit is defined with respect to a ring $(R, \oplus, \ominus, \otimes, 0, 1)$. In particular, it uses gates that correspond to $\oplus, \ominus, \otimes : R \times R \to R$. Such a circuit gives rise to a function $f : \Sigma^n \to R$, where $n$ is the number of input nodes.

Central to our results will be the class $\mathsf{PNC}^1$, which corresponds to the class of languages $L$ for which there is a family $\mathcal{C}_1 = \{C_n\}_{n \geq 0}$ of log-depth arithmetic circuits with $C_n$ represents a function with $n$ input nodes such that, for all $w \in \Sigma^n$, we have $w \in L$ iff $C_n(w) > 0$ (*positivity* check). If we replace the positivity check $C_n(w) > 0$ with the zero or *equality* check $C_n(w) = 0$, we obtain the class $\mathsf{ENC}^1$.[3] The following relations are known between these classes (assuming some level of uniformity for the circuit

---

[3]While this class is often denoted as $\mathsf{C}_{=}\mathsf{NC}^1$, we use the simplified notation $\mathsf{ENC}^1$ for succinctness.

classes when compared with L or P):

$$TC^0 \subseteq NC^1 \subseteq ENC^1 \subseteq PNC^1 \subseteq L$$
$$\subseteq NC^2 \subseteq \cdots \subseteq NC \subseteq P.$$

Whether these relations are proper is a long-standing open question in complexity theory (just as the question of P vs. NP). That said, we will proceed as in complexity theory, namely, associate a neural model with one of the above complexity class by showing that it *captures a complete problem in the class* and is *subsumed by the class*. We define complete problems for these circuit classes in the standard way, i.e., under first-order (FO) reductions. For $NC^1$, natural complete problems include simulating a DFA (i.e., the word problem for $S_5$), multiplying matrices over the boolean semiring, and evaluating boolean formulas (Buss, 1987). For $PNC^1$, complete problems can be defined that, informally, extend these $NC^1$-complete problems with arithmetic values (cf. Proposition 3; Caussinus et al., 1998).

We note that $PNC^1$ is generally thought to be "just above" $NC^1$ in the sense that it can be simulated by NC circuits of depth $O(\log n \log^* n)$.[4] For sequence of length 64K to 1M, the overhead factor in parallel runtime relative to $NC^1$ is just 3. We will therefore think of the class $PNC^1$ as nearly as efficiently parallelizable as $NC^1$ in general and as transformers in particular.

## 3. Parallelizability Limits of Nonlinear RNNs

We start with a folklore upper bound on the expressivity of nonlinear RNNs (proof in Appendix A for completeness):

**Proposition 1** (Upper Bound). *Let $M$ be an RNN over $\mathbb{Q}$ with $s = O(\text{poly}(n))$ precision. Then the language recognized by $M$ is also recognizable by a Turing machine that uses $O(\max\{s, \log n\})$ space and $\text{poly}(n)$ time.*

**Corollary 1.** *Poly-precision RNNs can be simulated in P and log-precision RNNs can be simulated in L.*

This result gives a *fully sequential* simulation for poly-precision RNNs in P. On the other hand, since $L \subseteq NC^2$, log-precision nonlinear RNNs can be parallelized to depth $O(\log^2 n)$, incurring a factor of $O(\log n)$ in circuit depth compared to transformers. We will next show that both of these upper bounds are essentially tight, barring major breakthroughs in complexity theory. That is, poly-precision RNNs likely cannot be parallelized at all to poly-logarithmic depth, and, for log-precision RNNs, the $O(\log n)$ penalty relative to transformers likely cannot be removed.

---

[4]Mereghetti & Palano (2000) mention that the iterated matrix multiplication problem is in $NC^1$. However, they do not provide a proof or reference, and this turns out to be not quite accurate. The best known simulation of $PNC^1$ by NC circuits recursively uses the Chinese Remainder Theorem and incurs a small overhead of $O(\log^* n)$, requiring total depth $O(\log n \log^* n)$ (Jung, 1985).

### 3.1. Poly-Precision Nonlinear RNNs Are P-Complete

We first show that nonlinear RNNs with polynomial precision can simulate Turing machines, taking inspiration from the classical construction of Siegelmann & Sontag (1995). It will follow that they can solve P-complete problems, which means they cannot be parallelized (to poly-log depth) unless $NC = P$. Towards showing this, we prove that RNNs whose gating function is an MLP can simulate a multi-stack machine, which is Turing-complete (proof in Appendix A):

**Theorem 1** (Turing-Completeness). *For any multi-stack machine $M$, there exists a one-layer MLP RNN that recognizes the same language as $M$.*

*Proof sketch.* The idea follows Siegelmann & Sontag (1995). We use a poly-precision scalar in the RNN state to represent each stack and leverage the fact that a ReLU network (used in the recurrence) can update the stack by pushing and popping. □

Since a multi-stack machine is Turing complete, it follows from Theorem 1 that poly-precision nonlinear RNNs can solve a P-complete problem (proof in Appendix A):

**Corollary 2** (P-Completeness). *There is a one-layer MLP RNN whose language is P-complete under FO reductions.*

It follows that, under standard complexity conjectures, poly-precision nonlinear RNNs cannot be parallelized effectively:

**Corollary 3** (Poly-Precision Depth Lower Bound). *If $NC \neq P$, there is a nonlinear RNN that, for any $k$, cannot be simulated by an NC circuit family of depth $O(\log^k n)$.*

### 3.2. Log-Precision Nonlinear RNNs Are L-Complete

We have established that poly-precision nonlinear RNNs cannot be parallelized effectively assuming $NC \neq P$. However, in practice, LLMs use bounded precision, which motivates similarly analyzing nonlinear RNNs with log precision. Corollary 1 shows such RNNs are in L; we now show they can also solve an L-complete problem, establishing that they likely cannot be parallelized as effectively as transformers or linear RNNs under standard conjectures.

Prior work (Merrill & Sabharwal, 2025b) has used the *graph connectivity* problem as a prototypical L-complete problem when analyzing models such as log-depth transformers. This problem, however, does not naturally lend to a left-to-right single-pass solution that would be suitable for formalizing the additional expressive power of nonlinear RNNs. To work around this hurdle, we consider the following *sorted* and *deterministic* variant of the problem where each graph node has exactly one outgoing edge and the nodes are presented in a topologically sorted order:

**Definition 11.** *Sorted deterministic graph connectivity* is the problem that takes as input a source node $s$, sequence

of directed edges $(i_1, j_1), \ldots (i_n, j_n)$, and target node $t$, all encoded in unary. The edges are *deterministic* — meaning that each $i$ has at most one outgoing edge $(i, j)$ — and *topologically sorted* — meaning $i_1 < \cdots < i_n$. The output is 1 if and only if there is a path of edges from $s$ to $t$.

Perhaps surprisingly, this simpler variant of deterministic graph connectivity remains L-complete:

**Proposition 2.** *Sorted deterministic graph connectivity is* L-*complete under* FO-*reductions.*

We defer the proof to Appendix A.2. Intuitively, this is achieved by reduction from deterministic (but not necessarily sorted) graph connectivity. The latter is well-known to be L-complete (cf. Jones, 1975; Cook & McKenzie, 1987; Sipser, 1997), and is in fact also complete under FO-reductions just like the general graph connectivity problem (Immerman, 1998). The reduction makes $O(n)$ copies of the deterministic graph and connects the $i$-th copy to $(i+1)$-st copy. In this way, the resulting deterministic graph is polynomial in the size of the original graph and at the same time topologically sorted.

The topological sortedness condition allows a logspace machine to solve the sorted deterministic graph connectivity problem by reading the graph in one direction (from left to right). This is also the reason that a single-layer nonlinear RNN can solve the sorted deterministic graph connectivity problem. Since the indices of the vertices (given in unary) can be encoded in binary with $O(\log n)$ bits, nonlinear RNNs can solve this problem with only log precision.

**Theorem 2** (L-Complete). *There exists a one-layer log-precision MLP RNN that solves sorted deterministic graph connectivity, an* L-*complete problem.*

*Proof sketch.* The proof leverages the computational model of counter machines (Fischer et al., 1968; Weiss et al., 2018; Merrill, 2021), a generalization of finite automata augmented with a fixed number of integer-valued registers that has been heavily used to analyze RNNs (Weiss et al., 2018; Merrill, 2019). Lemma 2 shows a counter machine can solve sorted deterministic graph connectivity. Moreover, Lemma 3 shows a log-precision MLP RNN can simulate a counter machine.[5] Thus, we conclude that an MLP RNN can also solve sorted deterministic graph connectivity. □

It is a longstanding open problem (Allender, 2023) if any L-complete problem has a boolean circuit family of depth $o(\log^2 n)$. Thus, we have:

**Corollary 4** (Log-Precision Parallelism Limit). *Unless every problem in* L *admits an* NC *circuit family of* $o(\log^2 n)$

---

[5]While counter machines have been leveraged extensively to analyze RNNs in the past (Weiss et al., 2018; Merrill et al., 2020), Lemma 3 is novel and interesting in its own right.

*depth, there exists a log-precision nonlinear RNN that requires* NC *circuit families of depth* $\Omega(\log^2 n)$ *to simulate.*

Corollary 4 relates to recent work on parallelizing nonlinear RNNs via Newton's method, i.e., exactly computing the nonlinear RNN through iterated linear approximation (Danieli et al., 2026; Gonzalez et al., 2026). Gonzalez et al. (2025) show that this method runs in time $O(\log^2 n)$ for predictable dynamical systems, matching our theoretical expectation for parallel runtime of nonlinear RNNs.

## 4. Simulating LRNNs with Parallel Circuits

Having established the limits of parallelizing nonlinear RNNs, we turn to understanding the degree to which linear RNNs can be parallelized and, conversely, their limitations for expressing inherently sequential computation. Prior theoretical work has also studied the expressivity of linear RNNs from a circuit complexity perspective, revealing expressivity advantages of *some* linear RNNs over transformers (Merrill et al., 2024). In particular, whereas linear RNNs with simple parameterizations like S4 and Mamba fall into $\mathsf{TC}^0$, the same complexity class as transformers, more complex parameterizations like DeltaNet (Grazzi et al., 2025) and RWKV (Peng et al., 2025) can express $\mathsf{NC}^1$-complete finite state tracking problems. This past work suggests a key advantage of well-parameterized linear RNN architectures over transformers, but leaves open a precise understanding of whether this additional expressivity implies drawbacks in parallelism, i.e, additional overhead in the sequential runtime required to simulate LRNNs relative to transformers.

Transformers only compute functions in $\mathsf{TC}^0$, which implies they can be simulated by $\mathsf{NC}^1$ circuits of depth $O(\log n)$. It is known that linear RNNs can be parallelized to near log depth using the parallel SCAN algorithm (Blelloch, 1990), but this does not immediately yield a circuit complexity characterization comparable to that of transformers. We address this gap by showing that all linear RNNs, independent of parameterization details, can be simulated in the complexity class $\mathsf{PNC}^1$ of bounded fan-in log-depth circuits with arithmetic gates:

**Theorem 3** (LRNN Upper Bound). *The language recognized by any LRNN over* $\mathbb{Q}$ *is in* $\mathsf{PNC}^1$.

*Proof.* Since FO-uniform polynomial-size circuit families are closed under finite composition (Merrill & Sabharwal, 2025a, Proposition 3), the proof proceeds by simulating each component in the network in FO-uniform $\mathsf{PNC}^1$.

As a building block, we need gates that take as input bit representations for $(s, t)$ and output the integer-pair representation $(s, t)$ for the rational number $s/t$. This can be done by an FO-uniform poly-sized log-depth arithmetic $\mathbb{Z}$-circuit (here, the measurement is with respect to the num-

ber of bits of $s$ and $t$), as follows: if $s$ is represented by $b_{n-1} \ldots b_0 \in \{0,1\}^*$, then $s = \sum_{i=0}^{n-1} b_i \cdot 2^i$. Iterated sum, fast exponentiation, and multiplication by a constant can all be done by a uniform arithmetic $\mathbb{Z}$-circuit with depth $O(\log n)$ and size polynomial in $n$.

Next, we write the output in a *convolutional form* (Merrill & Sabharwal, 2023; Merrill et al., 2024):

$$\mathbf{y}_t = \mathbf{x}_t + \sum_{j=0}^{i} \mathbf{q}_t^\top \left( \prod_{k=0}^{j} \mathbf{A}_{j-k} \right) \mathbf{b}_{i-j}. \qquad (1)$$

By Lemma 1, there is an FO-uniform $\mathsf{TC}^0$, and hence $\mathsf{NC}^1$, circuit family that maps the inputs $\bar{x}$ to each $\mathbf{A}_t$ and $\mathbf{b}_t$. Using the previously described bits-to-rational conversion gates, we turn these into rational representations. Each of these matrices has bounded dimension $d$. Iterated matrix multiplication, as well as iterated matrix addition, for matrices of a bounded dimension over any semiring $\mathbb{K}$ (here $\mathbb{K} = \mathbb{Q}$) can be done using a log depth poly-sized $\mathbb{K}$-arithmetic circuit (Vollmer, 1999). Since $\mathbb{Q}$-arithmetic circuits can be simulated using $\mathbb{Z}$-circuits incurring only a constant number of operations for each gate (see Proposition 4), we obtain a $\mathsf{GapNC}^1$ circuit for outputting $\mathbf{w}^\top \mathbf{y}_t$. The last step is to perform a positivity check, $\mathbf{w}^\top \mathbf{y}_t > 0$, which puts the problem in $\mathsf{PNC}^1$. □

Theorem 3 applies generally to LRNNs over $\mathbb{Q}$ (i.e., poly-precision LRNNs). For the subclass of log-precision LRNNs, it is in fact possible to obtain a tighter characterization. Let $\mathsf{AC}^0[\mathsf{ENC}^1]$ denote languages that can be solved by boolean $\mathsf{AC}^0$ circuits that can make use (as a blackbox) of any boolean circuit that comes from an $\mathsf{ENC}^1$-arithmetic circuit. It is known (cf. Datta et al., 2012) that $\mathsf{ENC}^1 \subseteq \mathsf{AC}^0[\mathsf{ENC}^1] \subseteq \mathsf{PNC}^1$.

**Theorem 4** (Log-Precision LRNN Upper Bound)**.** *The language recognized by any log-precision LRNN over $\mathbb{Q}$ is in $\mathsf{AC}^0[\mathsf{ENC}^1]$.*

*Proof.* Let $C$ denote the FO-uniform $\mathsf{GapNC}^1$ circuit constructed in the proof of Theorem 3 that computes $\mathbf{w}^\top \mathbf{y}_t > 0$. Owing to log precision of the LRNN, this output can take only polynomially many possible values $v$. Thus, instead of performing a positivity check on $\mathbf{w}^\top \mathbf{y}_t$ (as in construction in Theorem 3), we leverage log-precision to simulate the positivity check with polynomially many *equality* checks. Specifically, for each possible $v$, we extend $C$ to build an FO-uniform $\mathsf{ENC}^1$-circuit $C_v$ that checks whether the output $\mathbf{w}^\top \mathbf{y}_t$ equals $v$. The positivity check is then simply an unbounded fan-in OR, and hence an $\mathsf{AC}^0$ circuit, whose inputs are the outputs of $C_v$ for $v > 0$. □

As noted in Section 2.5, $\mathsf{PNC}^1$ can be simulated by $O(\log n \log^* n)$ depth $\mathsf{NC}$ circuits. We therefore have:

**Corollary 5.** *The language recognized by any LRNN over $\mathbb{Q}$ is also recognized by an $\mathsf{NC}$ circuit family of depth $O(\log n \log^* n)$.*

It follows that LRNNs are nearly as **efficiently parallelizable** on hardware as transformers, paying only an $O(\log^* n)$ depth penalty beyond the $O(\log n)$ depth needed to implement transformers with $\mathsf{NC}$ circuits. On the other hand, we will show next that well-parameterized LRNNs significantly gain (over transformers) on the side of expressivity.

As a by-product of our above results, we also obtain the following boundary between linear RNNs and nonlinear RNNs. Recall that (sorted, deterministic) graph connectivity problem is $\mathsf{L}$-complete.

**Corollary 6.** *Assuming $\mathsf{L}$ does not admit $\mathsf{NC}$ circuits of depth $O(\log n \log^* n)$, LRNNs over $\mathbb{Q}$ cannot solve (sorted, deterministic) graph connectivity, but nonlinear RNNs can.*

It is only known that $\mathsf{L} \subseteq \mathsf{NC}^2$. It is still a long-standing open problem (Allender, 2023) whether any $\mathsf{L}$-complete problem admits $\mathsf{NC}$ circuits with depth $o(\log^2 n)$.

# 5. Expressivity Differences Between LRNNs

In Section 4, we showed all linear RNNs can only solve problems in $\mathsf{PNC}^1$. A natural question is whether this upper bound is tight, i.e., whether each linear RNN variant can solve hard problems in $\mathsf{PNC}^1$. For some variants like S4 and Mamba, we already know a tighter upper bound of $\mathsf{TC}^0$ can be given (Merrill et al., 2024), meaning these architectures do not achieve full expressivity in $\mathsf{PNC}^1$. However, several newer LRNN parameterizations of LRNNs that can express $\mathsf{NC}^1$-complete problems have been proposed: in particular several flavors of DLPR LRNNs as well as PD LRNNs (Section 2.3). We now analyze the expressivity of these different parameterizations, showing a conditional separation: while DPLR LRNNs attain $\mathsf{PNC}^1$-complete expressivity, we can show that PD LRNNs are contained within $\mathsf{NC}^1$.

## 5.1. DPLR LRNNs are $\mathsf{PNC}^1$-Complete

Two common popular DPLR LRNNs are RWKV-7 (Peng et al., 2025) and DeltaNet (Schlag et al., 2021). We will show both of these architectures can solve $\mathsf{PNC}^1$-complete problems, achieving the upper bound on LRNN expressivity from Section 4. We begin by defining a $\mathsf{PNC}^1$-complete problem, namely *iterated matrix multiplication*:

**Definition 12** (Iterated $3 \times 3$ Matrix Multiplication)**.** The input is a token stream of length $9n$ over $\{-1,0,1\}$, partitioned into $n$ consecutive blocks of length $9$. Block $t$ encodes a matrix $\mathbf{A}_t \in \mathbb{K}^{3 \times 3}$. The iterated matrix multiplication problem is to compute $\mathbf{P}_n = \prod_{t=1}^{n} \mathbf{A}_t \in \mathbb{Z}^{3 \times 3}$.

**Proposition 3** (Caussinus et al., 1998, Corollary 3.4)**.** *The $3 \times 3$ iterated multiplication problem is $\mathsf{GapNC}^1$-complete*

*under* FO *reductions, and thus checking if the* $(0,0)$ *entry of* $\mathbf{P}_n$ *is positive is* $\mathsf{PNC}^1$ *complete under* FO *reductions.*

We show RWKV-7 and DeltaNet can express this iterated matrix multiplication problem, establishing that they can solve $\mathsf{PNC}^1$-complete problems:

**Theorem 5** (DPLR $\mathsf{PNC}^1$-Completeness). *There exist a* 4*-layer RWKV-7 and a* 4*-layer DeltaNet that solves the iterated* $3 \times 3$ *matrix multiplication problem and, in particular, checks if the* $(0,0)$ *entry of* $\mathbf{P}_n$ *is positive.*

*Proof sketch.* For a full proof, see Appendix B.3 for RWKV-7 and Appendix B.5 for DeltaNet. The goal is to multiply a sequence of arbitrary matrices. If the transition matrices were allowed to have an arbitrary form, the construction would be easy with a single RNN layer. However, since the transition matrices are constrained to have DPLR form, we need a few extra layers to factor blocks of the given $3 \times 3$ matrices into blocks of DPLR matrices. To this end, we generalize an idea from Peng et al. (2025): use layers 1–3 to factor blocks of arbitrary matrices into DPLR matrices with the same product, and then multiply all of these DPLR matrices to get the overall product $\mathbf{P}_n$. $\square$

Theorem 5 shows that RWKV-7 and DeltaNet, two DPLR LRNNs, can indeed solve a $\mathsf{PNC}^1$-complete problem, providing a lower bound on their expressivity that matches the upper bound from Theorem 3. This however leaves open the possibility that even such LRNNs may not be able to solve much simpler problems within $\mathsf{PNC}^1$. To address this gap, we extend the argument to also derive a more *general lower bound*, stating that RWKV-7 can, in fact, solve *every* problem solvable by a *Weighted Finite Automaton (WFA)* over the semiring $\mathbb{Z}$ (WFA; see Definition 13 for a definition). WFA subsume the problem of iterated 3-by-3 matrix multiplications with matrices over $\{-1, 0, 1\}$, which we already remarked (see Exercise 5.10 of Vollmer, 1999 and Caussinus et al., 1998) to be $\mathsf{GapNC}^1$-complete. In other words, the set of functions $f : \Sigma^* \to \mathbb{Z}$ definable by WFAs contains a $\mathsf{GapNC}^1$-complete problem. Thus, considering "acceptance" by a WFA of an input string $w$ with the condition $f(w) > 0$, there is a $\mathsf{PNC}^1$-complete WFA language.

**Theorem 6** (DPLR WFA Simulation). *For any* $n$-*state WFA* $\mathcal{A}$ *over* $\mathbb{Q}$, *there exist a* 4*-layer RWKV-7 and a* 4*-layer DeltaNet that both compute* $f_{\mathcal{A}}(w)$.

A proof sketch and details are deferred to Appendix B.2 for RWKV-7 and to Appendix B.4 for DeltaNet.

### 5.2. PD LRNNs are $\mathsf{NC}^1$-Complete

PD LRNNs are another recently proposed LRNN variant that can succinctly represent the $\mathsf{NC}^1$-complete state tracking problem (Terzic et al., 2025). We analyze the expres-

sive power, showing that, in contrast to DPLR LRNNs, PD LRNNs are constrained to $\mathsf{NC}^1$:

**Theorem 7.** *Let* $M$ *be a multi-layer PD LRNN over* $\mathbb{Q}$. *Then the language recognized by* $M$ *is in* FO-*uniform* $\mathsf{NC}^1$.

*Proof Sketch.* The theorem reduces to showing that the product of a sequence of matrices $\mathbf{P}_1\mathbf{D}_1, \ldots, \mathbf{P}_n\mathbf{D}_n$ can be computed in FO-uniform $\mathsf{NC}^1$. Terzic et al. (2025) observe PD matrices are closed under multiplication, so this product also has PD form. Let $\prod_{i=1}^n \mathbf{P}_i\mathbf{D}_i = \tilde{\mathbf{P}}_n\tilde{\mathbf{D}}_n$. We derive a closed form for $\tilde{\mathbf{P}}_i$ and $\tilde{\mathbf{D}}_i$, and show these can be computed in uniform $\mathsf{NC}^1$; full proof in Appendix C. $\square$

Theorem 7 establishes a conditional expressivity gap between DPLR and PD LRNNs: while DPLR architectures like DeltaNet and RWKV-7 can express some $\mathsf{PNC}^1$-complete problems, PD LRNNs are restricted to $\mathsf{NC}^1$. Moreover, since it has previously been shown that PD LRNNs can recognize regular languages (Terzic et al., 2025), we see that they can solve $\mathsf{NC}^1$-complete problems as well.

In the one-layer case, we can show that deterministic WFAs can be simulated by PD LRNNS in the following sense:

**Theorem 8.** *Any language recognized by a deterministic WFA with a zero threshold over* $\mathbb{Q}$ *can also be recognized by a one-layer PD LRNN over* $\mathbb{Q}$.

Proof in Appendix D.2. From Theorems 7 and 8, we have:

**Corollary 7.** *Any language defined by a deterministic WFA with a zero threshold can be simulated in* $\mathsf{NC}^1$. *Moreover, there exists an* $\mathsf{NC}^1$-*complete language that can be recognized by a deterministic WFA with a zero threshold.*

## 6. Experiments

We have provided a concise theoretical characterization of the expressive strengths and limitations of (non)linear RNNs. We now empirically evaluate the learnable capacity of recurrent models on the Deterministic Graph Connectivity and Iterated $3 \times 3$ Matrix Multiplication. Specifically, we compare nonlinear RNNs with linear RNN models (DeltaNet and RWKV-7), the linear RNN variant Mamba, and include Transformers as an additional baseline.

**Experimental Setup.** To better evaluate learning capacity, we construct balanced datasets with length generalization. The train set contains 70K samples uniformly drawn from $N \in [1, 100]$ and the validation set contains 20K samples drawn from the same range. We evaluated on three test splits, each containing 10K samples, drawn from $[1, 100]$, $[101, 200]$, and $[201, 300]$. Here, $N$ denotes the number of nodes in the graph task and the size of the matrix in the matrix multiplication task.

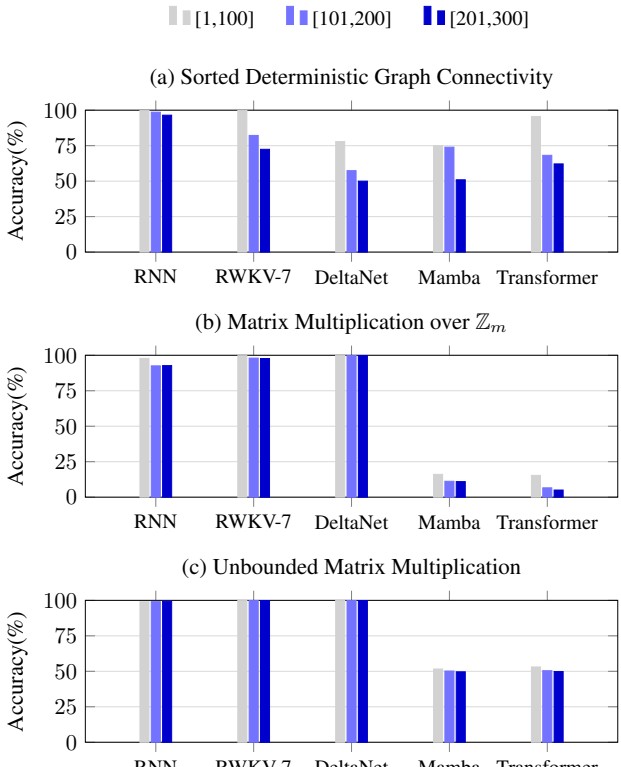

*Figure 2.* Accuracy across size/length ranges. (a) Deterministic graph connectivity over graph-size ranges. (b) Iterated $3 \times 3$ matrix multiplication over $\mathbb{Z}_m$. (c) Iterated $3 \times 3$ matrix multiplication over $\mathbb{Z}$. Models are evaluated on in-distribution ranges $[1, 100]$ and out-of-distribution ranges $[101, 200]$ and $[201, 300]$.

We used the nonlinear RNN and transformer (Sarrof et al., 2024), and linear models from Yang & Zhang (2024). All models are trained using the BCEWithLogitsLoss function. All models are trained using AdamW with learning rate $1e^{-4}$ and weight decay $10^{-4}$, batch size 64, gradient clipping at norm 1.0, and early stopping either when the accuracy of the in-distribution data reaches $100\%$ for three consecutive evaluations or when a maximum of 60K training steps are reached. Full benchmark details and training configurations are provided in the Appendix.

**Sorted Deterministic Graph Connectivity.** As shown in Figure 2 (a), all models achieve high in-distribution accuracy, indicating that sorted deterministic graph connectivity is learnable under matched training and test lengths. However, performance diverges markedly under length extrapolation. The nonlinear RNN maintains near-perfect accuracy across all bins, demonstrating strong length generalization. In contrast, Transformer, RWKV-7, Mamba, and DeltaNet exhibit substantial degradation as graph size increases, despite competitive performance. These results highlight the importance of recurrent inductive bias for length-generalizable algorithmic reasoning.

**Iterated Matrix Multiplication.** RWKV-7, the nonlinear RNN, and DeltaNet achieve near-perfect accuracy on the in-distribution data and degrade only moderately on the longer out-of-distribution sequences in (b). In contrast, Transformer and Mamba perform poorly across all lengths, suggesting limited capacity to capture the underlying algebraic structure even within the training regime. Architectural differences are more pronounced in the non-modular (c). RWKV-7, the nonlinear RNN, and DeltaNet retain perfect or near-perfect accuracy across both in-distribution and out-of-distribution lengths, despite unbounded integer growth. Mamba improves with increasing sequence length but remains substantially below the top-performing models, while the Transformer degrades sharply beyond training lengths. Overall, these results indicate that architectures with explicit recurrence or state-space structure are substantially better suited for iterated linear-algebraic computations than standard attention-based Transformers.

# 7. Conclusion

We have given a comprehensive characterization of the expressive power and parallelizability of different RNNs and LRNNs. First, we reveal fundamental expressivity gaps between linear and nonlinear RNNs, highlighting the deterministic graph connectivity problem as a simple separator. This circuit complexity perspective refines an older line of work on rational recurrences (Peng et al., 2018; Merrill et al., 2020), formalizing a fundamental tradeoff between the expressivity of the state update and the parallelizability of an RNN. Second, we compare the expressivity of different types of linear RNN. Whereas both PD and DPLR parameterizations have been shown to be capable of regular language recognition in prior work, our theory reveals fine-grained differences between these variants, with PD being $\text{NC}^1$-complete while DPLR LRNNs are $\text{PNC}^1$-complete. Experiments agree with our theoretical predictions about the expressivity differences between nonlinear RNNs and LRNNs and different LRNN variants.

Hybrid transformer–LRNN models have recently seen wide adoption (e.g., Waleffe et al., 2024; Kimi Team, 2025; Qwen Team, 2026) informed by emerging understanding of LRNN expressivity (Merrill et al., 2024; Grazzi et al., 2025; Peng et al., 2025; Merrill et al., 2026). Thus, we hope our theory can guide the design of LRNN and hybrid architectures that effectively balance expressivity and parallelism. Our results also suggest recent methods to parallelize *nonlinear* RNNs (Danieli et al., 2026; Gonzalez et al., 2025) are one pathway to more expressive hybrid models, though nonlinear RNNs must fundamentally pay a $\Theta(\log n)$ cost in parallel runtime under standard conjectures (cf. Corollary 4). It is an open question whether the additional expressivity of nonlinear RNNs justifies this overhead in practice.

## Impact Statement

This paper presents foundational work whose goal is to advance the field of machine learning. There are many potential societal consequences of our work, none of which we feel must be specifically highlighted here.

## Acknowledgments

WM thanks Mehryar Mohri for insightful discussions related to RNNs and WFAs as well as Brian Kitano and Aleksandar Terzić for finding issues in early drafts.

This material is based in part upon work supported by the European Union (ERC, LASD, 101089343, `https://doi.org/10.3030/101089343`).

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

# A. Nonlinear RNNs

**Proposition 1** (Upper Bound). *Let $M$ be an RNN over $\mathbb{Q}$ with $s = O(\text{poly}(n))$ precision. Then the language recognized by $M$ is also recognizable by a Turing machine that uses $O(\max\{s, \log n\})$ space and $\text{poly}(n)$ time.*

*Proof.* Given a log-precision RNN of depth $d$, we will simulate it left-to-right using a Turing machine $T$. Let $s' = \max\{s, \log n\}$. At each step $t$, for each layer, $T$ stores the hidden state $h_t$ at that layer using $s$ bits. Given $h_{t-1}$ and input $x_t$, by Lemma 1, $h_t$ can be computed in FO-uniform $\text{TC}^0$, which is contained in L. Thus, $T$ can compute $h_t$ using $O(\max\{s, \log n\})$ space[6] and $\text{poly}(n)$ time. This space can be reused for computing $h_{t+1}$, $h_{t+2}$, etc. Overall, the Turing machine uses $d \cdot O(\max\{s, \log n\})$ bits of storage space and runs in $\text{poly}(n)$ time, as desired. $\square$

## A.1. Polynomial Precision

**Theorem 1** (Turing-Completeness). *For any multi-stack machine $M$, there exists a one-layer MLP RNN that recognizes the same language as $M$.*

*Proof.* The idea is the same as the proof of Lemma 3, except that we will simulate an automaton with stacks of binary values rather than counters. The stacks allow update operations $\text{push}_0$, $\text{push}_1$, pop, and noop. Additionally, it has a head, which is the top element. We view each stack $s$ as a positive number $s \in [0, 2]$ that starts off as $s = 1$ and is modified using the following update operations:

$$\text{head}(s) = \mathbb{1}[s \geq 1]$$
$$\text{push}_v(s) = v + \frac{1}{2}s$$
$$\text{pop}(s) = \begin{cases} 2(s - \text{head}(s)) & \text{if } s \neq 1 \\ \text{noop}(s) & \text{otherwise} \end{cases}$$
$$\text{noop}(s) = s.$$

Moreover, we will let $s_t \in [0, 2]^k$ denote the vector of stacks at time $t$. The update operation $u_t \in \{\text{head}, \text{push}, \text{pop}, \text{noop}\}$ is determined via a finite lookup table $u$ as a function of $q_t$, $\sigma_t$, and $\text{head}(s_t)$. The state transition $\delta$ depends on the same inputs.

The implementation of these operations with a ReLU network follows the same idea as the proof of Lemma 3. We first construct a ReLU network to compute $\text{head}(s)$ from $s$ as a threshold with $\epsilon = 1/3$ (Yang et al., 2026, Section 4.7). Next, we compute $q_t = \delta(q_{t-1}, \sigma_t, \text{head}(s_t))$ and $u_t = u(q_{t-1}, \sigma_t, \text{head}(s_{t-1}))$ as finite lookup tables (Yang et al., 2026, Section 4.8). At this point, we have the correct next state, and it just remains to be shown that we can update the stacks. Let $\text{pop}_1(s) = 2s - 2$ and $\text{pop}_0(s) = 2s$. We compute the possible updated stacks $\text{push}_v(s_{t-1})$, $\text{pop}_v(s_{t-1})$, and $\text{noop}(s_{t-1})$ with parallel ReLU networks, using the fact that each is a linear function. For each stack $[s_t]_i$, we then compute its new value from these $u(s_{t-1})$'s using a selector gadget based on $[u_t]_i$, $\text{head}(s_{t-1})_i$, and whether $s = 1$ (Yang et al., 2026, Section 4.9). Thus, we can correctly compute $q_t$ and $s_t$ at each step. $\square$

**Corollary 2** (P-Completeness). *There is a one-layer MLP RNN whose language is P-complete under FO reductions.*

*Proof.* We leverage the same idea as Barcelo et al. (2026, Section 4.8). Let $L$ be any P-complete language under FO reductions. By construction $L$ can be recognized by a poly-time Turing machine and thus also by a multi-stack machine running in time $O(n^c)$ for some $c$. Define a new padded language $L' = \{w\square^{|w|^c} \mid w \in L\}$. Then $L'$ is P-complete and can be solved by a multi-stack machine, and thus also by an MLP RNN by Theorem 1. $\square$

## A.2. Logarithmic Precision

**Proposition 2.** *Sorted deterministic graph connectivity is L-complete under FO-reductions.*

*Proof.* We describe a simple FO reduction from deterministic (but not necessarily sorted) graph connectivity. The latter is well-known to be L-complete (cf. Jones, 1975; Cook & McKenzie, 1987; Sipser, 1997), and is in fact also complete under

---

[6]Here $\log n$ space is for intermediate computation of the L machine, and $s$ space is for its output.

FO-reductions just like the general graph connectivity problem (Immerman, 1998). We are given a deterministic graph $G$ with source $s = 1$, target $t = n$, and edges

$$(i_1, j_1), \ldots, (i_m, j_m)$$

with $i_k, j_k \in \{1, \ldots, n\}$, for each $k = 1, \ldots, m$, and $i_1 = s$ and $j_m = t = n$. We create a new deterministic graph $G'$ with edges

$$(i_1 + hn, j_1 + (h+1)n), \ldots, (i_m + hn, j_m + (h+1)n)$$

for each $h = 0, \ldots, m - 1$. We also add an additional node $v_F = nm + 1$ and edges $(n + hn, v_F)$. Note that $n$ has no outgoing edge in $G$, so $G'$ remains deterministic. That is, $v_0, \ldots, v_s$ is a path in $G$ iff $v_0, v_1 + n, v_2 + 2n, \ldots, v_s + sn$ is a path in $G'$. Thus, $t$ is reachable from $s$ in $G$ iff $v_F$ is reachable from $s$ in $G'$.

The above reduction can easily be seen to be implementable by a logspace-uniform $\mathsf{AC}^0$ reduction. The values $hn$ (for each $h = 1, \ldots, m$) can be precomputed by a logspace machine since this depends only on the input size (more precisely, one can enumerate enough of $hn$ for up to $m = O(n)$). The $\mathsf{AC}^0$ circuit simply needs to perform simple additions. $\qquad\square$

**Theorem 2** (L-Complete). *There exists a one-layer log-precision MLP RNN that solves sorted deterministic graph connectivity, an* L-*complete problem.*

Our proof of this theorem will leverage the computational model of counter machines (CMs; Fischer et al., 1968; Weiss et al., 2018; Merrill, 2021). Also called counter automata, CMs are a generalization of finite automata augmented with a fixed number of integer-valued registers. CMs been heavily used to analyze the expressive power of RNNs (Weiss et al., 2018; Merrill, 2019). Lemma 2 shows a counter machine can solve sorted deterministic graph connectivity. Moreover, Lemma 3 shows a log-precision RNN with a ReLU MLP gating function can simulate a counter machine. Thus, we conclude that a log-precision RNN with ReLU MLP gating can also solve sorted deterministic graph connectivity.

**Lemma 2.** *Sorted deterministic graph connectivity can be solved by counter machine.*

*Proof.* Fix a unary encoding of an instance

$$(s, (i_1, j_1), \ldots, (i_n, j_n), t)$$

with separators between $s$, each $i_m$, each $j_m$, and $t$. We assume that vertices are numbered in a topological order and encoded in unary by their numbers, so for every edge $(i_m, j_m)$ we have $j_m \geq i_m$. The machine uses three counters $S, I, T$ and $O(1)$ states to maintain the invariant that, after $k$ edges, $S$ encodes the furthest state reachable from $s$.

The first step is to process the initial block $0^s$. The machine starts in a state $q_s$ and increments $S$ on each symbol (leaving $I, T$ at 0), so after this block $S = s$.

We next enter a state $q_{\text{edge}}$ and process each edge $(i_m, j_m)$ in turn, maintaining the invariant that after $m$ edges we have $S = s_m$ (the node reached from $s$ after the first $m$ edges) and $I = 0$. For edge $(i_m, j_m)$, in a state $q_i$ the machine reads the unary block $0^{i_m}$ and on each symbol performs

$$S \leftarrow S - 1, \qquad I \leftarrow I + 1.$$

At the following separator it tests whether $S = 0$ using the zero-mask. If $S = 0$ (i.e. $s_{m-1} = i_m$), it moves to a state $q_j^=$ and on each symbol of the $j_m$-block simply does $S \leftarrow S + 1$ (leaving $I$ unchanged); since $S$ starts at 0, after $0^{j_m}$ we have $S = j_m$, and at the next separator it resets $I \leftarrow 0$ and returns to $q_{\text{edge}}$, so $s_m = j_m$. If instead $S \neq 0$, the machine moves to a state $q_j^{\neq}$. On each symbol of the $j_m$-block it now performs

$$S \leftarrow S + 1, \; I \leftarrow I - 1 \quad \text{if } I \neq 0, \qquad \text{and does nothing if } I = 0,$$

where the choice again uses only the zero-mask bit of $I$. Initially $I = i_m$, so after $\min\{i_m, j_m\}$ symbols we have added that many units back to $S$ and subtracted them from $I$. Because $j_m \geq i_m$, by the end of the $j_m$-block we have $I = 0$ and

$$S = (s_{m-1} - i_m) + i_m = s_{m-1},$$

and at the next separator we return to $q_{\text{edge}}$ with $S = s_m = s_{m-1}$ and $I = 0$. Thus the invariant is preserved across every edge.

After the last edge the machine enters a state $q_t$ with $S = s_n$ and $I = 0$. On the target block $0^t$ it updates

$$S \leftarrow S - 1, \qquad T \leftarrow T + 1$$

on each symbol, so at the end $S = s_n - t$. On the end-of-input marker it moves to an accepting state iff the zero-mask bit of $S$ is 0, i.e. iff $S = 0$, which is equivalent to $s_n = t$. □

**Lemma 3.** *Let $M$ be a real-time counter machine. There exists a one-layer MLP RNN with log precision that recognizes the same language as $M$.*

*Proof.* We divide partition the hidden state $h_t$ into three parts: $s_t$, which encodes a finite state as a one-hot vector, $c_t^+$, which encodes the value of positive counters, and $c_t^-$, which encodes the absolute value of negative counters. Define $q_t$ as the state encoded by $s_t$ and $c_t = c_t^+ - c_t^-$. We will show how to construct an MLP $f$ such that, applying $h_t = f(h_{t-1}, x_t)$, we will have $s_t, c_t^+, c_t^-$ such that

$$q_t = \delta(q_{t-1}, \sigma_t, z(c_{t-1}))$$
$$c_t = u(q_{t-1}, \sigma_t, z(c_{t-1}))(c_{t-1}).$$

We first construct a ReLU network to compute $z(c_{t-1})$ as a function of $c_{t-1}^+$ and $c_{t-1}^-$ using the standard construction for equality checks with a tolerance of $\epsilon = 1/3$ (Yang et al., 2026, Section 4.7). We then construct ReLU networks to compute $\delta$ and $u$ (with output encoded as one-hot vectors) given $q_{t-1}, \sigma_t, z(c_{t-1})$, which is possible since both are finite lookup tables (Yang et al., 2026, Section 4.8). We compute $z$ with $\delta$ to get a one-hot encoding of $q_t = \delta(q_{t-1}, \sigma_t, z(c_{t-1}))$, i.e., our updated state vector $s_t$. Similarly, we compose $z$ with $u$ to get a one-hot encoding of the update function $u_t = u(q_{t-1}, \sigma_t, z(c_{t-1}))$ to apply to the counters. The new values of $c_t^+$ and $c_t^-$ can be obtained by a ReLU network that computes several continuous piecewise-linear functions of $c_{t-1}^+, c_{t-1}^-$ in parallel for different updates $u \in \{\times 0, +0, +1, -1\}$ (Yang et al., 2026, Section 4.1) and then applies a conditional selector for $u = u_t$ (Yang et al., 2026, Section 4.9).

We can then accept or reject a string of length $n$ based on $s_n, c_n^+, c_n^-$ using an MLP. The construction uses log precision because $q_t$ is in a finite set and $c_t^+, c_t^- = O(n)$ . □

# B. Lower Bounds for DPLR LRNNS

This section contains four simulation results, organized by architecture and target computation:

- RWKV-7 simulates WFAs over $\mathbb{Q}$ (Subsection B.2).

- RWKV-7 computes iterated $3 \times 3$ matrix multiplication over $\mathbb{Q}$ (Subsection B.3).

- DeltaNet simulates WFAs over $\mathbb{Q}$ (Subsection B.4).

- DeltaNet computes iterated $3 \times 3$ matrix multiplication over $\mathbb{Q}$ (Subsection B.5).

## B.1. Preliminaries

**Definition 13** (WFA Path Form). Let $\langle \mathbb{K}, \oplus, \otimes \rangle$ be a semiring. A weighted finite automaton (WFA) is a tuple $\mathcal{A} = (Q, \Sigma, \{M_\sigma\}_{\sigma \in \Sigma}, \alpha, \omega)$ where $Q = \{1, \ldots, n\}$ is a finite state set, each $M_\sigma \in \mathbb{K}^{n \times n}$ is a transition matrix, and $\alpha, \omega \in \mathbb{K}^n$ are initial/final weight vectors, respectively. The WFA assigns a score to a string $w = w_1 \cdots w_n \in \Sigma^*$ via

$$f_{\mathcal{A}}(w) = \bigoplus_{\pi = \{(s_k, t_k)\}_{k=1}^n} \alpha_{\pi_1} \otimes \left( \bigotimes_{i=1}^n [M_{w_i}]_{\pi_k, \pi_{k+1}} \right) \otimes \omega_{\pi_n},$$

where $\pi$ iterates over all paths of states of length $n$.

Equivalently, defining matrix multiplication over $\langle \mathbb{K}, \oplus, \otimes \rangle$, the computation of a WFA can be defined according to

$$f_{\mathcal{A}}(w) = \alpha^\top \otimes M_{w_1} \otimes \ldots \otimes M_{w_n} \otimes \omega \in \mathbb{K}.$$

**Definition 14** (Streaming iterated $3 \times 3$ matrix multiplication over $\mathbb{Z}$)**.** The input is a token stream of length $9N$ over alphabet $\mathbb{Z}$, partitioned into $N$ consecutive blocks of length 9. The $t$-th block encodes a matrix $A^{(t)} \in \mathbb{Z}^{3 \times 3}$ in row-major order:

$$\left( A_{1,1}^{(t)}, A_{1,2}^{(t)}, A_{1,3}^{(t)}, \ A_{2,1}^{(t)}, A_{2,2}^{(t)}, A_{2,3}^{(t)}, \ A_{3,1}^{(t)}, A_{3,2}^{(t)}, A_{3,3}^{(t)} \right).$$

The task is to output the product

$$P_N := A^{(1)} A^{(2)} \cdots A^{(N)} \in \mathbb{Z}^{3 \times 3} \qquad \text{(equivalently, its 9 entries).}$$

## B.2. Simulating WFAs with RWKV-7

Let $\mathbf{e}_i \in \mathbb{K}^d$ denote the $i$th standard basis *column* vector.

**Definition 15** (RWKV-style transition matrix (Peng et al., 2025))**.** Fix a dimension $d$. Given $w, a, \kappa \in \mathbb{K}^d$ and a scalar $\lambda \in \mathbb{K}$, define

$$A(w, a, \kappa; \lambda) := \mathrm{diag}(w) - \lambda \kappa \, (a \odot \kappa)^\top,$$

where $\odot$ is elementwise product.

To handle WFA transition matrices, we use a primitive that overwrites one coordinate by a dot product of the current state against a prescribed coefficient vector.

**Definition 16** (Dot-product overwrite matrix)**.** Fix a dimension $d$, an index $\mathrm{dst} \in \{1, \dots, d\}$, and a coefficient vector $c \in \mathbb{K}^d$ with $c_\mathrm{dst} = 0$. Define

$$U(\mathrm{dst}; c) := I - \mathbf{e}_\mathrm{dst} \mathbf{e}_\mathrm{dst}^\top + c \, \mathbf{e}_\mathrm{dst}^\top \in \mathbb{K}^{d \times d}.$$

Equivalently, $U(\mathrm{dst}; c)$ is the identity matrix with its $\mathrm{dst}$-th column replaced by $c$.

**Lemma 4** (Effect on row vectors)**.** *For any row vector $r \in \mathbb{K}^{1 \times d}$,*

$$(r \, U(\mathrm{dst}; c))_\mathrm{dst} = \sum_{i=1}^d r_i c_i, \qquad (r \, U(\mathrm{dst}; c))_j = r_j \ \ (j \neq \mathrm{dst}).$$

**Lemma 5** (Dot-product overwrite as an RWKV transition)**.** *Fix $d$, $\mathrm{dst}$, and $c \in \mathbb{K}^d$ with $c_\mathrm{dst} = 0$. Set*

$$w = \mathbf{1}, \qquad a = \mathbf{e}_\mathrm{dst}, \qquad \kappa = \mathbf{e}_\mathrm{dst} - c, \qquad \lambda = 1.$$

*Then $A(w, a, \kappa; 1) = U(\mathrm{dst}; c)$.*

*Proof.* Since $a = \mathbf{e}_\mathrm{dst}$, we have $a \odot \kappa = \kappa_\mathrm{dst} \mathbf{e}_\mathrm{dst}$. Because $c_\mathrm{dst} = 0$, $\kappa_\mathrm{dst} = 1$, so $a \odot \kappa = \mathbf{e}_\mathrm{dst}$. Thus

$$A = I - \kappa \, \mathbf{e}_\mathrm{dst}^\top = I - (\mathbf{e}_\mathrm{dst} - c) \mathbf{e}_\mathrm{dst}^\top = I - \mathbf{e}_\mathrm{dst} \mathbf{e}_\mathrm{dst}^\top + c \, \mathbf{e}_\mathrm{dst}^\top = U(\mathrm{dst}; c). \qquad \square$$

We now show that any $n \times n$ matrix can be applied to a row vector using $\Theta(n)$ overwrite steps, provided we have $n$ additional scratch coordinates.

**Lemma 6** (Apply an arbitrary $n \times n$ matrix using $2n$ overwrites)**.** *Let $P \in \mathbb{K}^{n \times n}$. Consider row vectors in dimension $2n$ written as $[x \mid s]$ with $x, s \in \mathbb{K}^{1 \times n}$. There exist $m := 2n$ overwrite matrices $G_1(P), \dots, G_m(P) \in \mathbb{K}^{2n \times 2n}$ such that for all $x, s$,*

$$[x \mid s] \, G_1(P) \cdots G_m(P) \ = \ [xP \mid xP].$$

*Proof. Phase 1 (compute $xP$ into scratch).* For each $j \in \{1, \dots, n\}$, define $c^{(j)} \in \mathbb{K}^{2n}$ by $c_i^{(j)} = P_{i,j}$ for $1 \leq i \leq n$ and $c_i^{(j)} = 0$ for $n+1 \leq i \leq 2n$. Let $G_j(P) := U(n+j; c^{(j)})$. By Lemma 4, this overwrites scratch coordinate $n+j$ with $\sum_{i=1}^n x_i P_{i,j} = (xP)_j$, leaving all other coordinates unchanged. After $n$ steps, $[x \mid s] G_1(P) \cdots G_n(P) = [x \mid xP]$.

*Phase 2 (copy scratch back to main).* For each $j \in \{1, \dots, n\}$ let $d^{(j)} := \mathbf{e}_{n+j} \in \mathbb{K}^{2n}$ and set $G_{n+j}(P) := U(j; d^{(j)})$. This overwrites main coordinate $j$ with the dot product against $\mathbf{e}_{n+j}$, i.e. the current scratch coordinate $n+j = (xP)_j$, leaving scratch unchanged. Thus after $n$ more steps, $[x \mid xP] G_{n+1}(P) \cdots G_{2n}(P) = [xP \mid xP]$. $\square$

B.2.1. SIMULATION THEOREM

**Theorem 9** (RWKV-7 simulates weighted finite automata over $\mathbb{Q}$). *Fix $\mathbb{K} = \mathbb{Q}$. Let $\mathcal{A} = (Q, \Sigma, \{M_\sigma\}, \alpha, \omega)$ be an $n$-state WFA over $\mathbb{K}$ (Definition 13), and let $w = w_1 \cdots w_T$ be an input word. Then there exists a 4-layer RWKV-7 network whose output at position $T$ equals $f_\mathcal{A}(w) = \alpha M_{w_1} \cdots M_{w_T} \omega \in \mathbb{Q}$.*

*Proof.* We use the "3-layer router + 1-layer multiplicative simulator" template of Peng et al. (2025) (Appendix D), instantiated with a block length $m := 2n$ and arithmetic dimension $2n$.

**Block parameters.** Let $m := 2n$. Write each position as $t = \ell m + \tau$ with $\ell \geq 0$ and $1 \leq \tau \leq m$. Introduce a padding token $\perp$ and set $w_t := \perp$ for all $t \leq 0$, with $M_\perp := I$. Let the *previous block* string be

$$\tilde{w}^{(\ell)} := w_{(\ell-1)m+1} \cdots w_{\ell m} \in (\Sigma \cup \{\perp\})^m,$$

which is well-defined for all $\ell$ due to padding.

**Offline factorization of block products.** For each length-$m$ block string $\tilde{w} = \tilde{w}_1 \cdots \tilde{w}_m$, define the block product

$$P(\tilde{w}) := M_{\tilde{w}_1} \cdots M_{\tilde{w}_m} \in \mathbb{K}^{n \times n}.$$

Apply Lemma 6 to obtain overwrite matrices $G_{\tilde{w},1}, \ldots, G_{\tilde{w},m} \in \mathbb{K}^{2n \times 2n}$ such that for all $x, s \in \mathbb{K}^{1 \times n}$,

$$[x \mid s] \prod_{i=1}^{m} G_{\tilde{w},i} = [xP(\tilde{w}) \mid xP(\tilde{w})].$$

Fix one such factorization for each $\tilde{w}$.

**Router (layers 1–3).** Define a lookup table $\Xi$ whose key is

$$\left( t \bmod 2m, \; w_t, w_{t-1}, \ldots, w_{t-(2m-1)} \right),$$

and whose outputs are:

1. the next factor $G_{\tilde{w}^{(\ell)},\tau}$ to apply at time $t$ (encoded so that the arithmetic layer can set $(w, a, \kappa)$ accordingly, using Lemma 5);

2. a "completion" readout vector $\hat{\omega}_t$ (defined below).

Both depend only on the key: $\tilde{w}^{(\ell)}$ is contained in the last $2m$ tokens whenever $\ell \geq 1$ (and is all padding when $\ell = 0$), and $\tau$ is determined by $t \bmod 2m$. By the finite-window lookup construction of Peng et al. (2025) (Appendix D; instantiated with block length $m$), a 3-layer RWKV-7 can compute $\Xi$ at every position and write its outputs into the residual stream.

**Arithmetic layer (layer 4).** Use a single wkv head of dimension $2n$. Initialize the first row of the wkv matrix state using a single additive update on the first token: set $v_1 = \mathbf{e}_1$ and $\tilde{k}_1 = [\alpha \mid \alpha]$, so the state becomes $\mathbf{e}_1^\top \tilde{k}_1$. For all $t \geq 2$, set the additive term to 0 (i.e. $v_t = \tilde{k}_t = \mathbf{0}$), so evolution is purely multiplicative: $S_t = S_{t-1} A_t$.

At each time $t$, the router specifies the factor $G_{\tilde{w}^{(\ell)},\tau}$. Choose RWKV transition parameters so that $A_t = G_{\tilde{w}^{(\ell)},\tau}$; since each factor is an overwrite matrix, this is possible by Lemma 5.

**Completion vector and correctness.** Let $p_t$ denote the first row of the arithmetic layer's wkv state $S_t$. As in Peng et al. (2025) (Appendix D), one checks by induction that for $t = \ell m + \tau$,

$$p_t = [\alpha \mid \alpha] \left( \prod_{b=0}^{\ell-2} \prod_{i=1}^{m} G_{\tilde{w}^{(b)},i} \right) \left( \prod_{i=1}^{\tau} G_{\tilde{w}^{(\ell-1)},i} \right),$$

with the convention that empty products are $I$.

Define the within-block prefix product

$$R_t := M_{w_{\ell m+1}} \cdots M_{w_t}, \qquad v_t := R_t \, \omega \in \mathbb{K}^{n \times 1}.$$

Now define the completion vector:

$$\hat{\omega}_t := \left( \prod_{i=\tau+1}^{m} G_{\tilde{w}^{(\ell-1)},i} \right) \cdot \begin{bmatrix} v_t \\ \mathbf{0} \end{bmatrix} \in \mathbb{K}^{2n \times 1}.$$

This $\hat{\omega}_t$ is a fixed function of the router key (position mod $2m$ and last $2m$ tokens), hence it can be returned by the lookup table $\Xi$.

Finally, compute the scalar readout $p_t \hat{\omega}_t$:

$$p_t \hat{\omega}_t = \alpha \, M_{w_1} \cdots M_{w_t} \, \omega$$

using Lemma 6 on each completed block. In particular, at $t = T$ the RWKV-7 output equals $f_{\mathcal{A}}(w)$. $\qquad \square$

### B.3. Iterated $3 \times 3$ Matrix Multiplication with RWKV-7

**Theorem 10** (RWKV-7 computes iterated $3 \times 3$ matrix products over $\mathbb{Q}$). *Fix $\mathbb{K} = \mathbb{Q}$. There exists a 4-layer RWKV-7 network such that, given the length-$9N$ stream encoding $A^{(1)}, \ldots, A^{(N)}$, the network outputs the 9 entries of $P_N := A^{(1)} A^{(2)} \cdots A^{(N)}$ (equivalently, $\mathrm{vec}(P_N) \in \mathbb{Q}^9$) at the final position $t = 9N$.*

*Proof sketch; full proof follows later.* Layers 1–2 compute the position modulo 18. This yields (i) the *within-block offset* $\tau \in \{1, \ldots, 9\}$ and (ii) a *block-parity bit* $b \in \{0,1\}$ that we use to alternate between two halves of an arithmetic state. Layer 3 stores the last 18 tokens (two consecutive length-9 blocks) and, from the key $(t \bmod 18, \ w_t, \ldots, w_{t-17})$, look up the RWKV transition parameters needed at time $t$. Concretely, at each token it outputs (in the residual stream) the *destination index* and *coefficient vector* for a single dot-product overwrite update implementing one coordinate of the map $x \mapsto x \, B(A^{(\ell-1)})$, where $A^{(\ell-1)}$ is the *previous* block's matrix. (We treat the pre-sequence padding as "block 0" encoding $A^{(0)} := I_3$.) Layer 4 maintains an 18-dimensional row state $[x^{(0)} \mid x^{(1)}] \in \mathbb{Q}^{1 \times 18}$ inside the first row of a wkv matrix state, where each half has length 9. During each length-9 block, layer 4 applies *nine* dot-product overwrites that compute a new vector $x_{\mathrm{new}} = x_{\mathrm{old}} B(A^{(\ell-1)})$ in the *inactive* half, using the parity bit $b$ to select which half is source vs. destination. At the final position $t = 9N$, the destination half contains $\mathrm{vec}(A^{(1)} \cdots A^{(N-1)})$. A *completion readout* (computed by the router from the final finite-window key, which contains the entire last block and hence determines $A^{(N)}$) outputs the 9 coordinates of $\mathrm{vec}(A^{(1)} \cdots A^{(N)})$ via nine dot products. $\qquad \square$

*Proof.* We follow the same "router (layers 1–3) + multiplicative arithmetic (layer 4)" template used in the RWKV-7 regular-language construction (Appendix D of Peng et al. (2025)).

**Step 1: reduce $3 \times 3$ matrix multiplication to a 9-dimensional linear recurrence.** Let $\mathrm{vec} : \mathbb{Q}^{3 \times 3} \to \mathbb{Q}^{1 \times 9}$ be row-major vectorization:

$$\mathrm{vec}(P) := (P_{1,1}, P_{1,2}, P_{1,3}, P_{2,1}, P_{2,2}, P_{2,3}, P_{3,1}, P_{3,2}, P_{3,3}).$$

Define the *block-diagonal embedding*

$$B(A) := \begin{bmatrix} A & 0 & 0 \\ 0 & A & 0 \\ 0 & 0 & A \end{bmatrix} \in \mathbb{Q}^{9 \times 9}.$$

Right-multiplication by $A$ acts independently on the three rows of $P$, hence

$$\mathrm{vec}(PA) = \mathrm{vec}(P) \, B(A) \qquad \text{for all } P, A \in \mathbb{Q}^{3 \times 3}. \tag{2}$$

Let $P_t := A^{(1)} \cdots A^{(t)}$ and $x_t := \mathrm{vec}(P_t)$. Then

$$x_0 = \mathrm{vec}(I_3), \qquad x_t = x_{t-1} B(A^{(t)}).$$

Thus it suffices to maintain the 9-dimensional state $x_t$ and repeatedly apply the linear map $x \mapsto x B(A)$.

**Step 2: implement** $x \mapsto xB(A)$ **with dot-product overwrites in dimension** 18. We use the dot-product overwrite primitive: for any destination coordinate dst and coefficient vector $c$ with $c_{\mathrm{dst}} = 0$, the overwrite matrix $U(\mathrm{dst}; c)$ replaces exactly one coordinate of a row vector with a dot product (Lemmas 4 and 5). In particular, each overwrite matrix can be realized as a *single* RWKV multiplicative transition (Lemma 5).

We maintain an arithmetic row state in dimension 18 written as

$$p = [x^{(0)} \mid x^{(1)}], \qquad x^{(0)}, x^{(1)} \in \mathbb{Q}^{1 \times 9},$$

together with a bit $b \in \{0, 1\}$ indicating the *active* half $x^{(b)}$ (source for dot products). Let $\pi(i, j) := 3(i - 1) + j$ be the row-major index map for $i, j \in \{1, 2, 3\}$.

Fix a $3 \times 3$ matrix $A$. The coordinates of $xB(A)$ are length-3 dot products:

$$(xB(A))_{\pi(i,j)} = \sum_{k=1}^{3} x_{\pi(i,k)} A_{k,j}. \tag{3}$$

We compute these 9 values into the inactive half $x^{(1-b)}$ using 9 overwrite steps. For each pair $(i, j) \in \{1, 2, 3\}^2$, define:

- *Destination coordinate* in the inactive half:

$$\mathrm{dst}(b; i, j) := 9(1 - b) + \pi(i, j) \in \{1, \ldots, 18\}.$$

- *Coefficient vector* $c^{(b;i,j)}(A) \in \mathbb{Q}^{18}$ supported only on the three active coordinates corresponding to row $i$:

$$c^{(b;i,j)}(A)_{9b+\pi(i,k)} := A_{k,j} \quad (k = 1, 2, 3), \qquad c^{(b;i,j)}(A)_u := 0 \text{ otherwise.}$$

Since $\mathrm{dst}(b; i, j)$ lies in the *inactive* half, we have $c^{(b;i,j)}(A)_{\mathrm{dst}(b;i,j)} = 0$, so the overwrite is well-formed. Applying $U(\mathrm{dst}(b; i, j); c^{(b;i,j)}(A))$ overwrites precisely the destination coordinate with the dot product of the current row state against $c^{(b;i,j)}(A)$ (Lemma 4), which equals the right-hand side of (3). Moreover, because each coefficient vector is supported only on the active half, these overwrites do not depend on (and hence do not interfere with) already-written coordinates in the inactive half.

If we apply these overwrites for all $(i, j)$ (in any order, e.g. row-major order), then the inactive half becomes $x^{(b)}B(A)$ while the active half remains unchanged:

$$[x^{(0)} \mid x^{(1)}] \longmapsto [x^{(0)} \mid x^{(1)}] \cdot \left( \prod_{(i,j)} U(\mathrm{dst}(b; i, j); c^{(b;i,j)}(A)) \right) = [x^{(0)} \mid x^{(1)}] \text{ with } x^{(1-b)} \leftarrow x^{(b)}B(A). \tag{4}$$

**Step 3: align the** 9 **overwrites with the token stream using a 3-layer router.** We now explain how layers 1–3 produce, at each time $t$, exactly the overwrite parameters needed by layer 4.

Partition the input stream into $N$ blocks of length 9. For convenience, define an additional "block 0" consisting of padding symbols before the sequence; the router will interpret this as encoding $I_3$ and we write $A^{(0)} := I_3$. Let $A^{(\ell)}$ be the matrix encoded by block $\ell$ for $\ell = 1, \ldots, N$. At any position $t$ in block $\ell$ (so $\ell \in \{1, \ldots, N\}$), the model will apply the update corresponding to the *previous* matrix $A^{(\ell-1)}$ (one-block delay).

**Computing** $(\tau, b)$ **from position.** Let $\tau \in \{1, \ldots, 9\}$ be the within-block offset and let $b \in \{0, 1\}$ be the parity bit indicating whether the active half is the first ($b = 0$) or second ($b = 1$) half. Both are determined by $t \bmod 18$: specifically, $\tau = 1 + ((t - 1) \bmod 9)$ and $b$ is the indicator of whether $t \bmod 18 \in \{10, \ldots, 18\}$. By Peng et al. (2025, Lemma 6) instantiated with $n = 9$, a 2-layer RWKV-7 can output $t \bmod 18$, hence can provide $\tau$ and $b$ to deeper layers.

**Recovering the previous block.** At time $t$ in block $\ell$, the last 18 tokens $(w_t, w_{t-1}, \ldots, w_{t-17})$ contain the entire previous block $\ell - 1$ (all 9 tokens encoding $A^{(\ell-1)}$), together with a prefix of the current block. Therefore, the tuple

$$\left( t \bmod 18, \ w_t, w_{t-1}, \ldots, w_{t-17} \right)$$

uniquely determines the matrix entries of $A^{(\ell-1)}$ and the current offset $\tau$. By Peng et al. (2025, Lemma 7) instantiated with $n = 9$, a 3-layer RWKV-7 can implement an arbitrary lookup table on this finite key and write its outputs into the residual stream.

**What the router outputs.** Define a lookup table $\Xi$ keyed by $(t \bmod 18, \ w_t, \ldots, w_{t-17})$ whose output at time $t$ is:

1. the destination index $\mathrm{dst}(b; i, j)$ and coefficient vector $c^{(b;i,j)}(A^{(\ell-1)})$ for the overwrite corresponding to $\tau = \pi(i, j)$; and

2. (optionally) the bit $b$ for the final readout.

The existence of such a lookup table is immediate because the key space is finite. By the previous paragraph, layers 1–3 can compute $\Xi$ at every position.

Finally, using the weight-preparation linear maps in the time-mixing block, layer 4 converts $(\mathrm{dst}, c)$ into RWKV transition parameters realizing the overwrite matrix $U(\mathrm{dst}; c)$ via Lemma 5.

**Step 4: arithmetic in layer 4 and correctness.** Layer 4 uses a single wkv head of head dimension 18. Initialize the first row of the wkv matrix state at the first token by setting

$$v_1 = \mathbf{e}_1, \qquad \tilde{k}_1 = [\mathrm{vec}(I_3) \mid \mathbf{0}] \in \mathbb{Q}^{1 \times 18},$$

so the wkv state becomes $\mathbf{e}_1^\top \tilde{k}_1$ and its first row equals $p_1 = [\mathrm{vec}(I_3) \mid \mathbf{0}]$. For all subsequent positions $t \geq 2$, set the additive term to 0 (i.e. $v_t = \tilde{k}_t = \mathbf{0}$), so the first row evolves purely multiplicatively as $p_t = p_{t-1} A_t$.

At each position $t$ in block $\ell$, the router provides the overwrite parameters corresponding to the matrix $A^{(\ell-1)}$ and the overwrite index $\tau$. By the discussion above, we choose $A_t = U(\mathrm{dst}; c)$ so that exactly one coordinate of $p_{t-1}$ is overwritten. After consuming all 9 tokens of block $\ell$, equation (4) implies that the inactive half has been filled with

$$x^{(1-b)} \ = \ x^{(b)} B(A^{(\ell-1)}).$$

By (2), this corresponds to right-multiplying the current $3 \times 3$ product by $A^{(\ell-1)}$.

By induction over blocks, after finishing block $\ell$ the destination half contains

$$\mathrm{vec}\Big( A^{(1)} A^{(2)} \cdots A^{(\ell-1)} \Big).$$

In particular, after finishing the final block $\ell = N$, the destination half contains $\mathrm{vec}(P_{N-1})$ where $P_{N-1} := A^{(1)} \cdots A^{(N-1)}$. We now remove the need for an additional padding block by applying the final multiplication by $A^{(N)}$ in the *readout* at the last token.

**Completion readout at $t = T = 9N$.** Let $T := 9N$. Let $p_T = [x_T^{(0)} \mid x_T^{(1)}] \in \mathbb{Q}^{1 \times 18}$ denote the arithmetic layer's row state at time $T$, and let $b_T \in \{0, 1\}$ be the block-parity bit computed from $T \bmod 18$. Define the destination-half index $h_T := 1 - b_T \in \{0, 1\}$ (equivalently, $h_T = N \bmod 2$). Then $x_T^{(h_T)} = \mathrm{vec}(P_{N-1})$.

Since $T$ is the last token of block $N$, the router key $(T \bmod 18, \ w_T, \ldots, w_{T-17})$ contains the entire block $N$ and hence determines $A^{(N)}$. For each $(i, j) \in \{1, 2, 3\}^2$, define the completion vector

$$\hat{\omega}_T^{(i,j)} := \sum_{k=1}^{3} A_{k,j}^{(N)} \, e_{9h_T + \pi(i,k)} \in \mathbb{Q}^{18}, \qquad \text{where } \pi(i, j) = 3(i-1) + j.$$

Then

$$p_T \hat{\omega}_T^{(i,j)} = \sum_{k=1}^{3} x_{T,\pi(i,k)}^{(h_T)} A_{k,j}^{(N)} = \mathrm{vec}(P_N)_{\pi(i,j)}.$$

Each $\hat{\omega}_T^{(i,j)}$ is a fixed function of the router key, so it can be returned by the finite-window lookup in layer 3 at time $T$. Using nine parallel readouts at the final position (one per $(i, j)$), the network outputs the nine scalars $p_T \hat{\omega}_T^{(i,j)}$ in row-major order, which equal $\mathrm{vec}(P_N)$. $\qquad \square$

### B.4. Simulating WFAs with DeltaNet

This subsection shows that a 4-layer stacked DeltaNet can simulate any weighted finite automaton over $\mathbb{Q}$. Throughout this subsection we fix $\mathbb{K} = \mathbb{Q}$ and interpret all arithmetic in $\mathbb{Q}$.

#### B.4.1. DELTANET MEMORY UPDATE AND THE MULTIPLICATIVE STEP MATRIX

We consider a DeltaNet layer with matrix state $S_t \in \mathbb{K}^{d \times d}$ updated by

$$S_t \;=\; S_{t-1}\big(I - \beta_t k_t k_t^{\top}\big) \;+\; \beta_t v_t k_t^{\top}, \tag{5}$$

where $\beta_t \in \mathbb{K}$ and $k_t, v_t \in \mathbb{K}^d$ are computed from the layer input $x_t$ (via linear maps / an MLP, as in standard stacked DeltaNet blocks).

We will repeatedly use the *multiplicative step matrix*

$$H(\beta, k) \;:=\; I - \beta k k^{\top} \in \mathbb{K}^{d \times d}. \tag{6}$$

When $v_t = \mathbf{0}$, the update is purely multiplicative:

$$S_t \;=\; S_{t-1} H(\beta_t, k_t).$$

#### B.4.2. ROUTER PRIMITIVES: POSITION MODULO AND TOKEN BUFFERING

**Notation.** Let $e_i$ denote the standard basis *column* vector. We assume the input stream contains an explicit BOS token at $t = 1$.

**Computing $t \bmod 2m$.** The router needs a cyclic write index $\tilde{t} \equiv t \pmod{2m}$ to address a $2m$-slot token buffer.

**Lemma 7** (DeltaNet can compute position modulo $2m$). *For any positive integer $m$, there exists a 2-layer stacked DeltaNet whose output includes a one-hot encoding of $\tilde{t} := t \bmod 2m$ at every position $t$.*

*Proof.* The construction mirrors a standard "two-reflections-make-a-rotation" argument. Layer 1 produces the parity bit of $t$ (and can also detect $t = 1$ via the BOS token). Layer 2 maintains a 2-dimensional continuous state that advances by a fixed angle every two steps: it alternates between two Householder reflections of the form $I - 2\kappa\kappa^{\top}$, which is exactly $H(\beta, k)$ with $\beta = 2$ and $k = \kappa$ (and $v_t = 0$). The state therefore cycles through $2m$ distinct values, one for each residue class modulo $2m$. A finite MLP on top of a bank of linear readouts can map these $2m$ discrete states to a one-hot encoding of $\tilde{t}$. $\square$

**Exact token buffering by column overwrite.** We store the last $2m$ tokens by overwriting one designated column per step.

**Lemma 8** (Exact column overwrite). *Fix $d$ and an index $\mathrm{dst} \in \{1, \dots, d\}$. If $\beta_t = 1$ and $k_t = e_{\mathrm{dst}}$, then*

$$S_t \;=\; S_{t-1}(I - e_{\mathrm{dst}} e_{\mathrm{dst}}^{\top}) + v_t e_{\mathrm{dst}}^{\top},$$

*so all columns of $S_{t-1}$ are unchanged except column $\mathrm{dst}$, which becomes exactly $v_t$.*

*Proof.* Right-multiplying by $(I - e_{\mathrm{dst}} e_{\mathrm{dst}}^{\top})$ zeros column $\mathrm{dst}$ and leaves all other columns unchanged; adding $v_t e_{\mathrm{dst}}^{\top}$ writes $v_t$ into that column. $\square$

**Buffer layout.** Let $L := 2m$. In the buffer layer, use a DeltaNet head dimension

$$d_{\mathrm{buf}} \;:=\; (|\Sigma| + 1) + L.$$

We interpret:

- the first $|\Sigma| + 1$ coordinates as a one-hot token space (including padding/BOS as needed);
- the last $L$ coordinates as buffer-slot selectors.

At time $t$, let $\tilde{t} \in \{1, \ldots, L\}$ be the residue class from Lemma 7. Write the current token into the $\tilde{t}$-th buffer slot by setting

$$\beta_t = 1, \qquad k_t = e_{(|\Sigma|+1)+\tilde{t}}, \qquad v_t = e_{w_t} \in \mathbb{K}^{d_{\text{buf}}}.$$

By Lemma 8, column $(|\Sigma| + 1) + \tilde{t}$ stores exactly the one-hot token vector $e_{w_t}$. Using $L$ read heads with fixed queries $q^{(j)} = e_{(|\Sigma|+1)+j}$, the layer can read out all $L$ stored tokens (each as a one-hot vector in the first $|\Sigma| + 1$ coordinates) into the residual stream.

**Finite lookup.** Given the one-hot encoding of $\tilde{t}$ and the last $2m$ tokens (read from the buffer), the key space is finite. Thus, an (exponentially wide) feedforward network can implement an arbitrary lookup table on this key and output: (i) the next arithmetic-step parameters $(\beta, k)$ for Layer 4, and (ii) a readout vector used to decode the desired scalar output.

### B.4.3. ARITHMETIC PRIMITIVES REALIZABLE BY MULTIPLICATIVE DELTANET STEPS

We now show that the multiplicative matrix $H(\beta, k) = I - \beta k k^\top$ can realize the linear operations needed to apply arbitrary matrices.

**Lemma 9** (Coordinate scaling / clearing). *Fix $j \in \{1, \ldots, d\}$ and $s \in \mathbb{K}$. Let $k = e_j$ and $\beta = 1 - s$. Then*

$$H(\beta, e_j) \;=\; I - (1-s)e_j e_j^\top$$

*scales coordinate $j$ by $s$ and leaves all other coordinates unchanged. In particular, $s = 0$ clears coordinate $j$.*

*Proof.* For any row vector $r \in \mathbb{K}^{1 \times d}$,

$$rH(\beta, e_j) = r - (1-s)(re_j)e_j^\top,$$

so $(rH)_j = r_j - (1-s)r_j = sr_j$ and $(rH)_\ell = r_\ell$ for $\ell \neq j$. $\qquad\square$

**Definition 17** (Unit transvection). For src $\neq$ dst define the (column) unit transvection

$$T(\text{src} \to \text{dst}) \;:=\; I + e_{\text{src}} e_{\text{dst}}^\top.$$

Right-multiplication by $T(\text{src} \to \text{dst})$ updates a row vector $r$ by $r_{\text{dst}} \leftarrow r_{\text{dst}} + r_{\text{src}}$ and leaves all other coordinates unchanged.

**Lemma 10** (Unit transvection as three multiplicative DeltaNet steps). *Assume $\frac{1}{2}, \frac{1}{3} \in \mathbb{K}$. Fix src $\neq$ dst and define (in dimension $d$)*

$$u := e_{\text{src}} + e_{\text{dst}}, \qquad w := e_{\text{src}} + 2e_{\text{dst}}.$$

*Then*

$$H(2, u)\, H\big(\tfrac{1}{2}, e_{\text{src}}\big)\, H\big(\tfrac{1}{3}, w\big) \;=\; T(\text{src} \to \text{dst}) \;=\; I + e_{\text{src}} e_{\text{dst}}^\top.$$

*Proof.* All three factors act as the identity outside the 2D subspace spanned by $e_{\text{src}}, e_{\text{dst}}$, so it suffices to verify the $2 \times 2$ restriction. In the ordered basis $(e_{\text{src}}, e_{\text{dst}})$, we have:

$$H(2, (1,1)) = \begin{bmatrix} -1 & -2 \\ -2 & -1 \end{bmatrix}, \quad H(\tfrac{1}{2}, (1,0)) = \begin{bmatrix} \frac{1}{2} & 0 \\ 0 & 1 \end{bmatrix}, \quad H(\tfrac{1}{3}, (1,2)) = \begin{bmatrix} \frac{2}{3} & -\frac{2}{3} \\ -\frac{2}{3} & -\frac{1}{3} \end{bmatrix}.$$

Multiplying gives

$$\begin{bmatrix} -1 & -2 \\ -2 & -1 \end{bmatrix} \begin{bmatrix} \frac{1}{2} & 0 \\ 0 & 1 \end{bmatrix} \begin{bmatrix} \frac{2}{3} & -\frac{2}{3} \\ -\frac{2}{3} & -\frac{1}{3} \end{bmatrix} = \begin{bmatrix} 1 & 1 \\ 0 & 1 \end{bmatrix} = I + e_{\text{src}} e_{\text{dst}}^\top.$$

Embedding back into the full $d$-dimensional space proves the claim. $\qquad\square$

**Lemma 11** (Scaled add via a temp register). *Assume $\frac{1}{2}, \frac{1}{3} \in \mathbb{K}$. Fix distinct indices src, dst, tmp and a scalar $\lambda \in \mathbb{K}$. There exists a length-8 sequence of multiplicative matrices of the form $H(\beta, k)$ such that for every row vector $r$ with $r_{\text{tmp}} = 0$, the resulting row vector $r'$ satisfies:*

$$r'_{\text{dst}} = r_{\text{dst}} + \lambda r_{\text{src}}, \qquad r'_j = r_j \; (j \neq \text{dst}), \qquad r'_{\text{tmp}} = 0.$$

*Proof.* Use the following four-stage program, each stage implementable by multiplicative DeltaNet steps:

1. Copy src into tmp by a unit transvection $T(\text{src} \to \text{tmp})$, implemented using 3 steps via Lemma 10.

2. Scale tmp by $\lambda$ using one coordinate scaling: apply $H(1 - \lambda, e_{\text{tmp}})$ (Lemma 9).

3. Add tmp into dst by a unit transvection $T(\text{tmp} \to \text{dst})$, implemented using 3 steps via Lemma 10.

4. Clear tmp by scaling it by 0: apply $H(1, e_{\text{tmp}})$ (Lemma 9).

Starting from $r_{\text{tmp}} = 0$, the net effect is $r_{\text{dst}} \leftarrow r_{\text{dst}} + \lambda r_{\text{src}}$ and tmp returns to 0. The total number of multiplicative steps is $3 + 1 + 3 + 1 = 8$. $\qquad\square$

**Applying an arbitrary matrix with scratch coordinates.** We now show that any $n \times n$ matrix can be applied to a row vector using $O(n^2)$ multiplicative DeltaNet steps, given $n$ scratch coordinates and one additional temp coordinate.

**Lemma 12** (Apply any $n \times n$ matrix with scratch in $O(n^2)$ multiplicative steps). *Assume* $\frac{1}{2}, \frac{1}{3} \in \mathbb{K}$. *Let* $P \in \mathbb{K}^{n \times n}$. *Consider row vectors in dimension* $2n + 1$ *written as*

$$[x \mid s \mid t], \qquad x, s \in \mathbb{K}^{1 \times n}, \; t \in \mathbb{K}.$$

*There exists an explicit sequence of*
$$m := 8n^2 + 5n + 1$$
*matrices* $G_1(P), \ldots, G_m(P) \in \mathbb{K}^{(2n+1) \times (2n+1)}$, *each of the form* $H(\beta, k)$, *such that for all* $x, s, t$,

$$[x \mid s \mid t]\, G_1(P) \cdots G_m(P) \;=\; [xP \mid xP \mid 0].$$

*Proof.* Let the coordinates be indexed as:

$$\text{main: } 1, \ldots, n, \quad \text{scratch: } n + 1, \ldots, 2n, \quad \text{temp: } 2n + 1.$$

**Phase 1 (clear scratch and temp).** For each scratch coordinate $n + j$ apply a clear:

$$H(1, e_{n+j}) \quad (j = 1, \ldots, n),$$

and clear temp by $H(1, e_{2n+1})$. This uses $n + 1$ steps and yields $[x \mid 0 \mid 0]$.

**Phase 2 (accumulate $xP$ into scratch).** For each destination $j \in \{1, \ldots, n\}$ and source $i \in \{1, \ldots, n\}$, update scratch coordinate $n + j$ by

$$(\text{scratch } j) \;\leftarrow\; (\text{scratch } j) + P_{i,j} \cdot (\text{main } i)$$

using Lemma 11 with

$$\text{src} = i, \qquad \text{dst} = n + j, \qquad \text{tmp} = 2n + 1, \qquad \lambda = P_{i,j}.$$

Each scaled add costs 8 multiplicative steps and leaves temp equal to 0. After all $n^2$ updates, scratch equals $xP$ while main remains $x$.

**Phase 3 (clear main).** Clear the main coordinates by applying $H(1, e_i)$ for $i = 1, \ldots, n$. This uses $n$ steps and yields $[0 \mid xP \mid 0]$.

**Phase 4 (copy scratch back to main).** For each $j = 1, \ldots, n$, add scratch coordinate $n + j$ into main coordinate $j$ using a unit transvection $T(n + j \to j)$, implemented by 3 multiplicative steps via Lemma 10. Since main is 0, after these $3n$ steps we have main $= xP$ and scratch remains $xP$.

Summing the step counts gives
$$(n + 1) + 8n^2 + n + 3n = 8n^2 + 5n + 1.$$

$\qquad\square$

B.4.4. SIMULATION THEOREM: A 4-LAYER DELTANET SIMULATES ANY WFA

**Theorem 11** (DeltaNet simulates weighted finite automata). *Fix $\mathbb{K} = \mathbb{Q}$. Let $\mathcal{A} = (Q, \Sigma, \{M_\sigma\}, \alpha, \omega)$ be an $n$-state WFA over $\mathbb{K}$ (Definition 13), and let $w = w_1 \cdots w_T$ be an input word. Then there exists a 4-layer stacked DeltaNet whose scalar output at every position $t$ equals*

$$\alpha \, M_{w_1} M_{w_2} \cdots M_{w_t} \, \omega \in \mathbb{K}.$$

*In particular, at $t = T$ the output equals $f_\mathcal{A}(w)$.*

*Proof sketch; full proof follows later.* We use the same finite-window *router* + *arithmetic* template as in Theorem 9 (compute $t \bmod 2m$, buffer the last $2m$ tokens, and look up the next arithmetic step), but implement these router components using DeltaNet primitives.

The arithmetic layer now uses only multiplicative DeltaNet steps $H(\beta, k) = I - \beta k k^\top$ (Definition 2.3) over $\mathbb{K} = \mathbb{Q}$, which guarantees $\frac{1}{2}, \frac{1}{3} \in \mathbb{K}$. Lemma 9 shows that $H(\beta, k)$ can implement coordinate scalings, and Lemma 10 shows that constant-length products of such steps implement unit transvections $r_{\mathrm{dst}} \leftarrow r_{\mathrm{dst}} + r_{\mathrm{src}}$. Combining these primitives, Lemma 12 gives an explicit program applying an arbitrary $n \times n$ matrix in $m = 8n^2 + 5n + 1$ multiplicative steps (using scratch space and one temporary coordinate). The router streams the corresponding $(\beta_t, k_t)$ parameters to the arithmetic layer exactly as in Theorem 9, and the same completion-vector decoding yields $f_\mathcal{A}(w)$. $\qquad\square$

*Proof.* Let

$$m \; := \; 8n^2 + 5n + 1$$

be the step budget from Lemma 12. For each position $t \geq 1$, write $t = \ell m + \tau$ with $\ell = \lfloor (t-1)/m \rfloor \geq 0$ and $1 \leq \tau \leq m$.

**Offline factorization of block products.** Introduce a padding symbol $\perp$ and set $w_t = \perp$ for $t \leq 0$ with $M_\perp := I$. For any length-$m$ block string $\tilde{w} = \tilde{w}_1 \cdots \tilde{w}_m \in (\Sigma \cup \{\perp\})^m$, define the block product

$$P(\tilde{w}) \; := \; M_{\tilde{w}_1} M_{\tilde{w}_2} \cdots M_{\tilde{w}_m} \in \mathbb{K}^{n \times n}.$$

Apply Lemma 12 to $P(\tilde{w})$ to obtain a fixed sequence

$$G_{\tilde{w},1}, \ldots, G_{\tilde{w},m} \in \mathbb{K}^{(2n+1)\times(2n+1)}, \qquad G_{\tilde{w},\tau} = H(\beta_{\tilde{w},\tau}, k_{\tilde{w},\tau}),$$

such that for all $[x \mid s \mid t]$,

$$[x \mid s \mid t] \prod_{\tau=1}^{m} G_{\tilde{w},\tau} \; = \; [xP(\tilde{w}) \mid xP(\tilde{w}) \mid 0].$$

Fix one such factorization for each possible $\tilde{w}$.

**Layer 1 (parity / first-position features).** Layer 1 produces a finite-valued encoding that distinguishes the first position (via BOS) and encodes parity. This can be done by a 1D multiplicative recurrence that toggles sign each step, together with the BOS token.

**Layer 2 (compute $t \bmod 2m$).** By Lemma 7, Layer 2 outputs a one-hot encoding of $\tilde{t} := t \bmod 2m$ in the residual stream.

**Layer 3 (buffer last $2m$ tokens and perform a finite lookup).** Set $L := 2m$ and build a $2m$-slot cyclic token buffer using Lemma 8, with the buffer layout described earlier. At time $t$, the buffer contains the last $2m$ tokens

$$(w_t, w_{t-1}, \ldots, w_{t-(2m-1)}),$$

and Layer 2 provides $\tilde{t} = t \bmod 2m$. Define a lookup table $\Xi$ whose key is

$$\left( \tilde{t}, \; w_t, w_{t-1}, \ldots, w_{t-(2m-1)} \right),$$

and whose outputs are:

1. the arithmetic parameters $(\beta_t^{(4)}, k_t^{(4)})$ specifying the matrix $G_{\tilde{w}^{(\ell-1)},\tau}$ to apply in Layer 4 (defined below);

2. a completion/readout vector $\hat{\omega}_t \in \mathbb{K}^{2n+1}$ (also defined below).

Since the key space is finite, an MLP can implement $\Xi$ exactly and write these outputs into the residual stream.

**Layer 4 (arithmetic: purely multiplicative DeltaNet simulation).** Layer 4 uses one DeltaNet head of dimension $2n+1$. We maintain only the *first row* of $S_t$ as the arithmetic row vector.

*Initialization.* Let $p_0 = [\alpha \mid 0 \mid 0] \in \mathbb{K}^{1 \times (2n+1)}$. On the first token, set $S_0 = 0$ and write $S_1 = e_1 p_0$ by choosing

$$\beta_1 = 1, \qquad v_1 = e_1, \qquad k_1 = p_0^\top.$$

Then $S_1 = \beta_1 v_1 k_1^\top = e_1 p_0$.

*Updates.* For all $t \geq 2$, set $v_t = \mathbf{0}$. Let $\tilde{w}^{(\ell-1)} := w_{(\ell-1)m+1} \cdots w_{\ell m}$ be the previous *completed* block string (using $\perp$ padding when $\ell = 0$), and let $\tau$ be the within-block offset of $t$. Layer 3 outputs $(\beta_t^{(4)}, k_t^{(4)})$ so that

$$H(\beta_t^{(4)}, k_t^{(4)}) \;=\; G_{\tilde{w}^{(\ell-1)},\tau}.$$

Thus, for $t \geq 2$ we have $S_t = S_{t-1} G_{\tilde{w}^{(\ell-1)},\tau}$. Let $p_t$ denote the first row of $S_t$; then $p_t = p_{t-1} G_{\tilde{w}^{(\ell-1)},\tau}$.

**Correctness via a completion vector.** Unrolling the recurrence, if $t = \ell m + \tau$ then

$$p_t = [\alpha \mid 0 \mid 0] \left( \prod_{b=0}^{\ell-2} \prod_{s=1}^{m} G_{\tilde{w}^{(b)},s} \right) \left( \prod_{s=1}^{\tau} G_{\tilde{w}^{(\ell-1)},s} \right),$$

with the convention that empty products are identity.

Define the current-block prefix product

$$R_t := M_{w_{\ell m + 1}} \cdots M_{w_t} \in \mathbb{K}^{n \times n}, \qquad v_t := R_t \, \omega \in \mathbb{K}^{n \times 1}.$$

Define the completion vector

$$\hat{\omega}_t := \left( \prod_{s=\tau+1}^{m} G_{\tilde{w}^{(\ell-1)},s} \right) \begin{bmatrix} v_t \\ \mathbf{0} \\ 0 \end{bmatrix} \in \mathbb{K}^{2n+1}.$$

This depends only on the router key (position mod $2m$ and the last $2m$ tokens), hence can be output by $\Xi$.

Now compute the scalar $p_t \hat{\omega}_t$: multiplying $p_t$ by the remaining factors $G_{\tilde{w}^{(\ell-1)},\tau+1} \cdots G_{\tilde{w}^{(\ell-1)},m}$ completes the application of the previous block product. By Lemma 12, the full block product $\prod_{s=1}^{m} G_{\tilde{w}^{(\ell-1)},s}$ maps the first $n$ coordinates $x$ to $x P(\tilde{w}^{(\ell-1)})$ (and duplicates it into scratch), so telescoping over completed blocks yields

$$p_t \hat{\omega}_t \;=\; \alpha \, M_{w_1} \cdots M_{w_t} \, \omega.$$

**Readout.** At position $t$, set a DeltaNet query vector $q_t := \hat{\omega}_t$ (routed from Layer 3) and read $o_t = S_t q_t$. Because $S_t = e_1 p_t$ has only the first row nonzero, we have $o_t = e_1 \cdot (p_t \hat{\omega}_t)$. An output head returns the first coordinate, which equals $\alpha M_{w_1} \cdots M_{w_t} \omega$. $\qquad\square$

## B.5. Iterated $3 \times 3$ Matrix Multiplication with DeltaNet

We now show that DeltaNet can solve the streaming iterated $3 \times 3$ matrix multiplication problem (Definition 14) over a bounded-precision modulus, using the same router+arithmetic template as Theorem 11. The main difference is that the arithmetic state dimension is constant ($n = 9$ after vectorization), so the required per-superblock step budget is a constant.

**Theorem 12** (DeltaNet computes iterated $3 \times 3$ matrix products over $\mathbb{Q}$). *Fix $\mathbb{K} = \mathbb{Q}$. There exists a 4-layer stacked DeltaNet such that, given the stream encoding $A^{(1)}, \ldots, A^{(N)} \in \mathbb{Q}^{3 \times 3}$ (Definition 14), the network outputs the 9 entries of*

$$P_N := A^{(1)} A^{(2)} \cdots A^{(N)} \in \mathbb{Q}^{3 \times 3}$$

*at the final position $t = 9N$.*

*Proof.* Work over $\mathbb{K} = \mathbb{Q}$.

**Step 1: reduce right-multiplication to a** $9$**-dimensional linear update.** Let $\text{vec} : \mathbb{Q}^{3\times3} \to \mathbb{Q}^{1\times9}$ be row-major vectorization. Define the block-diagonal embedding

$$B(A) := \begin{bmatrix} A & 0 & 0 \\ 0 & A & 0 \\ 0 & 0 & A \end{bmatrix} \in \mathbb{Q}^{9\times9}.$$

Then for all $P, A \in \mathbb{Q}^{3\times3}$,

$$\text{vec}(PA) = \text{vec}(P)\, B(A).$$

Let $x_t := \text{vec}(A^{(1)} \cdots A^{(t)})$. Then

$$x_0 = \text{vec}(I_3), \qquad x_t = x_{t-1}\, B(A^{(t)}).$$

Thus it suffices to maintain a 9-dimensional row state and repeatedly apply the linear map $x \mapsto xB(A)$.

**Step 2: choose a superblock length and step budget.** We will apply Lemma 12 with $n = 9$, i.e. we maintain an arithmetic state of dimension $2n + 1 = 19$ (main 9, scratch 9, temp 1), and can apply any $9 \times 9$ matrix in

$$m_{\text{arith}} := 8n^2 + 5n + 1 = 8 \cdot 81 + 45 + 1 = 694$$

multiplicative steps.

Each input matrix $A^{(t)}$ arrives as 9 tokens. We group matrices into superblocks of length

$$L := 78, \qquad \text{so each superblock has } 9L = 702 \text{ tokens.}$$

Since $702 \geq 694$, we can schedule the 694 arithmetic steps for a superblock and use the remaining $702 - 694 = 8$ token times as *identity* steps (i.e. apply $H(0, k) = I$).

We handle the one-superblock delay by a *completion readout* at the final position $T = 9N$, analogous to the RWKV completion-vector trick: the router computes a readout vector that (i) finishes the remaining multiplicative steps for the previous superblock and (ii) applies the final superblock product in the readout.

**Step 3: define superblock transition matrices.** Partition the matrix sequence into superblocks of length $L = 78$, except that the final superblock may be shorter. Let $B := \lceil N/L \rceil$ and let

$$r := N - (B-1)L \in \{1, \ldots, L\}.$$

For each superblock index $b < B$ define the full-superblock product

$$\Pi_b := B(A^{((b-1)L+1)}) \cdots B(A^{(bL)}) \in \mathbb{Q}^{9\times9}.$$

For the final superblock define the (possibly partial) product

$$\Pi_B := B(A^{((B-1)L+1)}) \cdots B(A^{(N)}) \in \mathbb{Q}^{9\times9}.$$

Then

$$\text{vec}(P_N) = x_0\, \Pi_1 \Pi_2 \cdots \Pi_B, \qquad x_0 = \text{vec}(I_3).$$

Partition the matrix sequence into superblocks of length $L = 78$, except that the final superblock may be shorter. Let $B := \lceil N/L \rceil$ and let

$$r := N - (B-1)L \in \{1, \ldots, L\}.$$

For each superblock index $b < B$ define the full-superblock product

$$\Pi_b := B(A^{((b-1)L+1)}) \cdots B(A^{(bL)}) \in \mathbb{Q}^{9\times9}.$$

For the final superblock define the (possibly partial) product

$$\Pi_B := B(A^{((B-1)L+1)}) \cdots B(A^{(N)}) \in \mathbb{Q}^{9\times9}.$$

**Step 4: offline factorization of each $\Pi_b$ into multiplicative DeltaNet steps.** By Lemma 12 (with $n = 9$), for each *full* superblock product $\Pi_b$ with $b < B$ there exists a length-694 sequence of $19 \times 19$ matrices of the form $H(\beta, k)$. Extend this to a length-702 sequence by appending 8 identity steps. Thus each full superblock $b < B$ is associated with a fixed sequence

$$G_{b,1}, \ldots, G_{b,702} \in \mathbb{Q}^{19 \times 19}, \qquad G_{b,\tau} = H(\beta_{b,\tau}, k_{b,\tau}).$$

(We also define $G_{0,\tau} := I$ for all $\tau$, corresponding to $\Pi_0 := I$.)

By Lemma 12 (with $n = 9$), for each $\Pi_b$ there exists a length-694 sequence of $19 \times 19$ matrices of the form $H(\beta, k)$ that maps

$$[x \mid s \mid t] \;\mapsto\; [x\Pi_b \mid x\Pi_b \mid 0].$$

Extend this to a length-702 sequence by appending 8 identity steps. Thus each superblock $b$ is associated with a fixed sequence

$$G_{b,1}, \ldots, G_{b,702} \in \mathbb{Q}^{19 \times 19}, \qquad G_{b,\tau} = H(\beta_{b,\tau}, k_{b,\tau})$$

such that after applying all 702 steps,

$$[x \mid s \mid t] \prod_{\tau=1}^{702} G_{b,\tau} \;=\; [x\Pi_b \mid x\Pi_b \mid 0].$$

Fix one such factorization for each possible superblock token string.

**Step 5: the 4-layer DeltaNet construction (router + arithmetic).** *Layers 1–2 (position modulo).* Let $m_{\text{tok}} := 702$ be the token length per superblock. Use Layers 1–2 to compute a one-hot encoding of $t \bmod (2m_{\text{tok}}) = t \bmod 1404$ (Lemma 7 with $m = m_{\text{tok}}$) and write it to the residual stream.

*Layer 3 (token buffer + lookup).* Maintain a cyclic buffer of the last $2m_{\text{tok}} = 1404$ tokens using the column overwrite mechanism (Lemma 8) with $L = 1404$ slots. At time $t$, the last 1404 tokens contain:

- the entire previous superblock (the last 702 tokens), encoding the $L = 78$ matrices whose product is $\Pi_{b-1}$,

- and a prefix of the current superblock.

Define a lookup table $\Xi$ keyed by

$$\left(t \bmod 1404, \; w_t, w_{t-1}, \ldots, w_{t-1403}\right)$$

whose output is the next arithmetic-step parameters $(\beta_t^{(4)}, k_t^{(4)})$ specifying the factor $G_{b-1,\tau}$ to apply at this time (one-superblock delay). Because the key space is finite (and of size at most $m^{1404} \leq N^{O(1)}$ under $m \leq N^c$), an MLP can implement $\Xi$ exactly.

*Layer 4 (arithmetic in dimension 19).* Layer 4 uses one DeltaNet head of dimension 19 and interprets the first row of $S_t$ as the arithmetic row vector $[x \mid s \mid t]$, where $x, s \in \mathbb{Q}^{1 \times 9}$ and $t \in \mathbb{Q}$.

**Initialization (first token).** Write $[\text{vec}(I_3) \mid 0 \mid 0]$ into the first row of $S_1$ using one additive write: set $\beta_1 = 1$, $v_1 = e_1$, and $k_1 = [\text{vec}(I_3) \mid 0 \mid 0]^\top$.

**Purely multiplicative evolution (all later tokens).** For all $t \geq 2$, set $v_t = \mathbf{0}$ so $S_t = S_{t-1} H(\beta_t, k_t)$. Layer 3 provides $(\beta_t, k_t)$ so that $H(\beta_t, k_t) = G_{b-1,\tau}$, i.e. the $\tau$-th factor for the previous superblock's product $\Pi_{b-1}$. During the first superblock, take $\Pi_0 = I$ and apply identity steps.

Let $T := 9N$ be the final position. Write

$$T = (B - 1)\, m_{\text{tok}} + \tau, \qquad m_{\text{tok}} := 702, \qquad 1 \leq \tau \leq m_{\text{tok}}.$$

(So $\tau = 9r$, where $r = N - (B-1)L$ is the number of matrices in the final superblock.) By construction, at time $T$ the arithmetic layer has applied the first $\tau$ factors of the previous full superblock $G_{B-1,1}, \ldots, G_{B-1,\tau}$ (or identity factors if $B = 1$), but it has *not* applied $\Pi_B$. We therefore decode $\text{vec}(P_N) = x_0 \Pi_1 \cdots \Pi_B$ using a completion readout at time $T$.

**Final readout via completion vectors.** Let $p_T \in \mathbb{Q}^{1 \times 19}$ denote the arithmetic layer's first-row state at time $T$. For each output coordinate $j \in \{1, \ldots, 9\}$ define the completion vector

$$\hat{\omega}_T^{(j)} := \left( \prod_{s=\tau+1}^{m_{\text{tok}}} G_{B-1,s} \right) \begin{bmatrix} \Pi_B e_j \\ \mathbf{0} \\ 0 \end{bmatrix} \in \mathbb{Q}^{19 \times 1},$$

where the product is interpreted as identity when $B = 1$ or $\tau = m_{\text{tok}}$. This $\hat{\omega}_T^{(j)}$ is a fixed function of the finite router key $(T \bmod 1404, \, w_T, w_{T-1}, \ldots, w_{T-1403})$, so it can be returned by the layer-3 lookup.

The network outputs the nine scalars

$$p_T \hat{\omega}_T^{(j)} = \left( \text{vec}(P_N) \right)_j, \qquad j = 1, \ldots, 9,$$

i.e. the 9 entries of $P_N$ in row-major order, at the final position $T = 9N$. $\qquad\qquad\square$

## C. Upper Bounds on PD LRNNs

**Theorem 7.** *Let $M$ be a multi-layer PD LRNN over $\mathbb{Q}$. Then the language recognized by $M$ is in* FO-*uniform* $\mathsf{NC}^1$.

*Proof.* Similar to the proof of Theorem 3, specifically the convolutional form of LRNNs in Equation (1), this proof reduces to showing that the product of a sequence $P_1 D_1, \ldots, P_n D_n$ of PD matrices can be computed in FO-uniform $\mathsf{NC}^1$. Terzic et al. (2025) observe PD matrices are closed under multiplication, so this product also has PD form. Let $\prod_{i=1}^n P_i D_i = \tilde{P}_n \tilde{D}_n$. We will provide a closed form for $\tilde{P}_i$ and $\tilde{D}_i$, and then argue that it can be easily computed using FO-uniform $\mathsf{NC}^1$ circuits.

Recall that $D_i$ is a diagonal matrix. Each $P_i$ is a column one-hot matrix in $\{0,1\}^{d \times d}$ and thus represents a function $\pi_i : \{1, 2, \ldots, d\} \to \{1, 2, \ldots, d\}$. When $P_i$ is full rank, $\pi_i$ represents a permutation of $d$ elements. In general, $\pi_i$ may map two inputs to the same output, i.e., collapse two elements into one. We will thus refer to it as a *relaxed permutation*.

We first observe that for any $d \times d$ matrix $A$, both $AP_i$ and $AD_i$ have a very simple form: $AP_i$ is the matrix obtained by applying the relaxed permutation $\pi_i$ to the columns of $A$, and $AD_i$ is the matrix obtained by scaling the $j$-th column of $A$ by the $j$-th diagonal entry of $D_i$. It follows that $AP_1 D_1, \ldots, P_n D_n$ is nothing but an alternating sequence of relaxed permutations and scaling of columns of $A$. Intuitively, as long as we adjust the scaling terms appropriately, we can switch the order, i.e., first apply all relaxed permutations to the columns of $A$ and then apply appropriate scaling; this, as we will shortly see, will make the matrix product easy to compute using uniform $\mathsf{NC}^1$ circuits. Formalizing this, we will indeed show that $\tilde{P}_n = \prod_{i=1}^n P_i$ and $\tilde{D}_n = \prod_{j=1}^n \tilde{\pi}_{n,j}(D_j)$, where $\tilde{\pi}_{n,j} = \prod_{i=n}^{j+1} \pi_i$ is the composition of the last $n - j$ relaxed permutations (by convention, $\tilde{\pi}_{n,n}$ is the identity permutation) and $\pi(D_j)$ denotes the relaxed permutation of the diagonal entries of $D_j$ according to $\pi$, leading to another diagonal matrix.

Conveniently, it is well-known that the product of $n$ fixed size matrices that take only finitely many values (in particular fixed size 0-1 matrices) can be computed in FO-uniform $\mathsf{NC}^1$ (Immerman & Landau, 1992). Thus $\tilde{P}_n$ can be computed using such circuits. The relaxed permutation $\tilde{\pi}_j$ can also be similarly computed, and applied to $D_j$. Lastly, $\tilde{D}_n$, being a product of $n$ diagonal matrices, consists of $n$ products of $n$ rationals each, which again can be computed in FO-uniform $\mathsf{NC}^1$.

All that remains to show is that $\tilde{P}_n = \prod_{i=1}^n P_i$ and $\tilde{D}_n = \prod_{j=1}^n \tilde{\pi}_{n,j}(D_j)$. We prove this by induction on $n$. The base case of $n = 1$ holds trivially. For $n \geq 2$, assume by induction that $\prod_{i=1}^{n-1} P_i D_i = \tilde{P}_{n-1} \tilde{D}_{n-1}$ where $\tilde{P}_{n-1} = \prod_{i=1}^{n-1} P_i$ and $\tilde{D}_{n-1} = \prod_{j=1}^{n-1} \tilde{\pi}_{n-1,j}(D_j)$. Then $\prod_{i=1}^n P_i D_i = \tilde{P}_{n-1} \tilde{D}_{n-1} P_n D_n$.

We will use a "swap" property of P and D matrices, as well as the column one-hot nature of the P matrices, to show that $\tilde{D}_{n-1} P_n D_n = P_n \tilde{D}_n$, which will finish the proof.

For this last part, observe that left-multiplying $P_n$ by the diagonal matrix $\tilde{D}_{n-1}$ amounts to scaling the $j$-th row of $P_n$ by the $j$-th diagonal entry of $\tilde{D}_{n-1}$. Since $P_n$ is column one-hot and thus has at most one non-zero entry in each column, this operation is the same as right-multiplying $P_n$ with a relax-permuted version of the diagonal matrix, namely $\pi_n(\tilde{D}_{n-1})$.

That is,

$$\tilde{D}_{n-1} P_n = P_n \, \pi_n(\tilde{D}_{n-1})$$

$$= P_n \, \pi_n \left( \prod_{j=1}^{n-1} \tilde{\pi}_{n-1,j}(D_j) \right)$$

$$= P_n \prod_{j=1}^{n-1} \pi_n \tilde{\pi}_{n-1,j}(D_j) \;=\; P_n \prod_{j=1}^{n-1} \tilde{\pi}_{n,j}(D_j)$$

Right-multiplying both sides by $D_n$, we obtain:

$$\tilde{D}_{n-1} P_n D_n = P_n \left( \prod_{j=1}^{n-1} \tilde{\pi}_{n,j}(D_j) \right) D_n = P_n \prod_{j=1}^{n} \tilde{\pi}_{n,j}(D_j) = P_n \tilde{D}_n$$

as desired, finishing the proof. $\qquad\square$

## D. Single-Layer LRNNs

### D.1. Inexpressibility of Deterministic Graph Connectivity

Our main results (Section 3) show that, assuming $\mathsf{PNC}^1 \neq \mathsf{L}$, multilayer linear RNNs cannot solve the sorted deterministic graph connectivity problem. The same result holds *unconditionally* in the single layer case, since a single-layer linear RNN is a WFA and no WFA can represent the sorted deterministic graph connectivity problem (cf. Section 3):

**Theorem 13.** *No weighted finite automaton can recognize sorted deterministic graph connectivity.*

*Proof.* Assume by way of contradiction that we can represent sorted deterministic graph connectivity. Then the Hankel matrix for this language has finite rank (Carlyle & Paz, 1971). But we can easily construct an unbounded-rank subblock of the Hankel matrix where the rows are $s$ and columns are $t$, and there are no edges in the graph (this yields the identity). Thus, we have reached a contradiction and this problem must not be representable. $\qquad\square$

A single-layer linear RNN can be simulated by a weighted finite automaton (Merrill et al., 2020, Theorem 4). Therefore, no linear RNN with just one layer can solve sorted deterministic graph connectivity, in the following sense:

**Corollary 8.** *The hidden state of a single linear RNN layer cannot solve sorted deterministic graph connectivity.*

The connection between LRNNs and WFAs shown here motivates considering a connection with an older line of work on rational recurrences (Peng et al., 2018). The central idea here, before the emergence of transformer, was that constraining RNNs so that their state was computable by a WFA was desirable because it made them more computationally efficient. Our results here reveal why: one-layer LRNNs are a kind of rational recurrence, and all rational recurrences can be computed in $\mathsf{PNC}^1$, i.e., in a highly parallel fashion. While stacking multiple LRNN layers may not remain a WFA (Merrill et al., 2020; Mohri, 2026), it will remain in $\mathsf{PNC}^1$ by Theorem 3.

### D.2. Single-Layer PD LRNNs and Deterministic WFAs

We say that a WFA (Definition 13) is *deterministic* if the initial state vector $\alpha$ and each column over every transition matrix $A_\sigma$ have at most one non-zero entry.

**Theorem 8.** *Any language recognized by a deterministic WFA with a zero threshold over $\mathbb{Q}$ can also be recognized by a one-layer PD LRNN over $\mathbb{Q}$.*

*Proof.* We can represent an arbitrary deterministic WFA transition matrix in PD form as follows. Let $d$ be the number of states. We set $b_t$ to be a one-hot encoding of the initial state at \$ and the zero vector elsewhere. For each $\sigma$ and state $q$, we set $P_\sigma$ to be 1 at the unique $q'$ to which $q$ transitions. We set the entry in $D_\sigma$ corresponding to $q$ to have the weight of this transition. Thus, a single-layer PD LRNN is equivalent to a deterministic WFA. $\qquad\square$

# E. Arithmetic Circuits over $\mathbb{Q}$

We define arithmetic circuits over the rational field $\langle \mathbb{Q}; +, -, \times, 0, 1 \rangle$, which we often write simply as $\mathbb{Q}$. We represent a rational number $a/b$ as a pair $(a, b)$ of integers. An arithmetic circuit $C$ over $\mathbb{Q}$ with $n$ input nodes gives rise to a function $f : \{0, 1\}^n \to \mathbb{Z} \times \mathbb{Z}$. That is, the circuit will output a representation $a/b$ of a rational number, but it might not be normalized. We show below that no problems arise in this way.

**Proposition 4.** *Any arithmetic circuit family $\mathcal{C} = \{C_n\}_{n \geq 1}$ over $\mathbb{Q}$ with logarithmic depth gives rise to a $\mathsf{GapNC}^1$ function. Therefore, it can be expressed as a boolean bounded-fan-in circuit family with $O(\log n \log^* n)$ depth.*

*Proof.* To prove this, simply note that over $\mathbb{Q}$:

$$\frac{a}{b} + \frac{c}{d} = \frac{ad + bc}{bc}$$

and

$$\frac{a}{b} \times \frac{c}{d} = \frac{ac}{bd}.$$

Similarly, $-\frac{a}{b} = \frac{-a}{b}$. Therefore, each $\mathbb{Q}$-operation can be simulated with at most four $\mathbb{Z}$-operations. Therefore, if $\mathcal{C}$ has log depth, then so are the $\mathbb{Z}$-circuits simulating $\mathcal{C}$.

So, as a corollary, $\mathcal{C}$ can be expressed as a boolean bounded-fan-in circuit family with $O(\log n \log^* n)$ depth. $\square$

# F. Additional Details for Experiments

**Benchmark.** We evaluate models on a suite of synthetic algorithmic tasks designed to test length generalization.

**Deterministic graph reachability.** We construct a synthetic reachability dataset over directed graphs with $n$ vertices. Vertices are assigned to two buckets under label-dependent constraints: for positive samples, vertices $0$ and $n$ lie in the same bucket, while for negative samples, they lie in different buckets. All remaining vertices are assigned independently with probability $p$. Directed edges are then added deterministically by connecting consecutive vertices within each bucket, yielding two disjoint directed paths. Each instance encodes a reachability query from vertex $0$ to vertex $n$ as a binary-serialized sequence.

**Iterated matrix multiplication with modulus.** This task requires computing prefix products of a sequence of $3 \times 3$ matrices over the finite ring $\mathbb{Z}_m$, where $m$ is a fixed prime. Each example has the form SOS $T \mid m \mid q_k \mid M_1 \mid \cdots \mid M_T$ EOS, where $T$ denotes the sequence length, $q_k$ is a fixed query index, and each matrix $M_t \in \{-1, 0, 1\}^{3 \times 3}$ is sampled independently and rejected until invertible modulo $m$. Let $P_t = M_1 \cdots M_t \bmod m$ denote the prefix product at step $t$. The target sequence is $v_1 \mid \cdots \mid v_T$, where $v_t$ equals the $q_k$-th entry of $P_t$ modulo $m$. Models are trained with a stepwise classification loss to predict $v_t$ at each step. The minimum sequence length for this task is $1$.

**Iterated matrix multiplication without modulus.** In this task, models multiply a sequence of $3 \times 3$ integer matrices over $\mathbb{Z}$ and predict a single binary label. Each example has the form SOS $T \mid M_1 \mid \cdots \mid M_T$ EOS, where $T$ is the number of matrices and each $M_t \in \{-1, 0, 1\}^{3 \times 3}$ is sampled i.i.d. with probabilities $(0.45, 0.10, 0.45)$ for entries $(-1, 0, 1)$. Let $P_T = M_1 M_2 \cdots M_T$ denote the product over $\mathbb{Z}$; the target label is defined as $y = 1$ iff $(P_T)_{0,0} = 0$, and $y = 0$ otherwise. We optionally construct balanced splits via rejection sampling to mitigate length-dependent artifacts. Since integer products may overflow for large $T$, labeling can be performed with an optional clipping cap that truncates intermediate values during multiplication. The minimum sequence length of this task is $1$.

Across both deterministic graph connectivity and iterated matrix multiplication, we use a largely standardized training setup to enable fair comparison across architectures. As shown in Table 1, all models are trained with AdamW, a fixed batch size of 128, and gradient clipping at 1.0. Learning rates and weight decay are kept consistent within each task, with minor adjustments for architectures that require stronger regularization, such as DeltaNet. Training is capped at 60k steps for deterministic graph connectivity and 60k steps for iterated matrix multiplication, ensuring comparable optimization budgets while isolating architectural inductive biases as the primary source of performance differences.

| Task | Model | Configuration | LR | Steps |
|------|-------|---------------|-----|-------|
| DGC | RNN | $d_m$=256, $L$=2, $d_h = 256$, | $3\times10^{-4}$ | 60k |
| | Transformer | $d_m$=256, $L$=2, $h = 4$ | $3\times10^{-4}$ | 60k |
| | Mamba | $d_m$=256, $L$=2, $d_{\text{state}}$=128 | $3\times10^{-4}$ | 60k |
| | RWKV-7 | $d_m$=256, $L$=2, $h = 4$ | $3\times10^{-4}$ | 60k |
| | DeltaNet | $d_m$=256, $L$=2, $h = 4$ | $3\times10^{-4}$ | 60k |
| IMM | RNN | $d_m$=256, $L$=2, $d_h = 256$, | $3\times10^{-4}$ | 60k |
| | Transformer | $d_m$=256, $L$=2, $h = 4$ | $3\times10^{-4}$ | 60k |
| | Mamba | $d_m$=256, $L$=2, $d_{\text{state}}$=64 | $3\times10^{-4}$ | 60k |
| | RWKV-7 | $d_m$=256, $L$=2, $h = 4$ | $3\times10^{-4}$ | 60k |
| | DeltaNet | $d_m$=256, $L$=2, $h = 4$ | $3\times10^{-4}$ | 60k |

*Table 1.* Training hyperparameters for deterministic graph connectivity (DGC) and iterated matrix multiplication (IMM). $d_m$, $L$, $d_h$, $h$, $d_{\text{state}}$, and $LR$ denote the model dimension, number of layers, hidden size, number of attention heads, SSM state expansion factor, and learning rate, respectively. All models are trained with AdamW, batch size 128, and gradient clipping at 1.0.

