# OpenReview forum: "Why Are Linear RNNs More Parallelizable?"
_ICML.cc/2026/Conference — ICML 2026 regular_

### Official Review · Reviewer_zax9 · 2026-03-09

**Soundness:** 4
**Presentation:** 4
**Significance:** 3
**Originality:** 4
**Overall Recommendation:** 5
**Confidence:** 3

**Summary:**

The paper provides a comprehensive characterization of the expressive power and parallelizability of various RNN and LRNN architectures. The theoretical analysis reveals a fundamental tradeoff between expressivity and parallelism in linear versus nonlinear recurrent neural networks. The paper also compare the expressivity among different type of LRNN.  The empirical results align with the theoretical predictions and support the identified expressivity differences between nonlinear RNNs, LRNNs, and various LRNN variants.

**Compliance With Llm Reviewing Policy:**

Affirmed.

**Final Justification:**

This is a paper in good shape, and I will maintain my current score.

**Key Questions For Authors:**

1) Based on Section 5 and Appendix C, the expressivity hierarchy between different LRNN variants (e.g., PD and DPLR) depends on specific parameterizations of the transition matrices. How robust are these results to architectural modifications?

2) Why not include PD LRNN as a baseline in Section 6? This will make the empirical study more comprehension.

3) I think this paper provides some valuable insights into the design of LLM architectures. My suggestion is that the author could  further discuss the potential impact of this work.

**Limitations:**

Yes

**Strengths And Weaknesses:**

Strengths:

1)	The main paper writing is clear and easy to follow. The appendix provides sufficient complementary information for both theorem proof and empirical study setting.

2)	The author provides a solid theoretical analysis of parallelizability of different RNNs and LRNNs. The assumptions are explicitly stated, and the proof steps are well organized and easy to follow.

3)	The empirical study is closely aligned with the theory, and the result can demonstrate the validity of author’s statement.

Weaknesses:

1)	The empirical evaluation is not fully comprehensive. For example, although Section 5.2 identifies PD LRNNs as an important component of the theoretical analysis, Section 6 does not include PD LRNNs as a baseline in the experiments.

2)	The empirical evaluation relies on relatively limited models and datasets. It remains unclear whether the observed patterns would generalize to larger-scale settings.

---

> ### Author Rebuttal · Authors · 2026-03-30
>
> Thank you for your review!
>
> Regarding empirical evaluation, our main goal was to provide some validation of the theoretical findings, which we perceive as the main contribution of this work.  The models and datasets chosen are not extensive but, we believe, still provide useful evidence that our expressivity analysis has practical implications on models trained to solve challenging tasks.
>
> In particular, as noted in our reply to Reviewer 2jXy, we did not do experiments with PD LRNNs because they are not implemented in the [Flash Linear Attention](https://github.com/fla-org/flash-linear-attention) repository that we used to replicate each architecture. We agree that it would be interesting to run experiments with PD LRNNs but found it somewhat painful in practice, and therefore we chose to prioritize other parts of the paper.
>
> Regarding the robustness of the expressivity results to specific parameterization choices, we note that we have abstracted away many specific details of RWKV and DeltaNet to the DPLR structure, which is the core shared property that enables the construction. Once we have a DPLR update that can have both positive and negative eigenvalues, the construction is general across both architectures, using fairly general purpose ideas. The PD construction similarly relies on the PD property of the transition matrix rather than specific paramerization details (e.g., the choice of nonlinearity). We focus on these classes of transition matrices because they encompass existing methods, but it is possible that other interesting “near-diagonal” parameterizations could be developed in the future (perhaps guided by theoretical analysis).
>
> Regarding potential impact, linear RNNs and hybrid models have recently gained wide practical adoption, though precise details of the underlying architecture are still somewhat open. For instance, whereas some large-scale releases (Qwen 3.5, Kimi Linear, Olmo Hybrid) use linear RNNs like GatedDeltaNet, others recent architecture work (e.g., xLSTM and M2RNN) re-incorporates nonlinear RNN components or explores methods for parallelizing nonlinear RNNs [1]. Our results help reveal the fundamental tradeoff between these approaches, and we hope they could help guide the convergence of these directions in the long term, similar to how past theoretical work on linear RNN expressiveness [2, 3] helped inspire innovations in the transition matrices for modern “slightly non-diagonal” linear RNNs. If accepted, we will revise the introduction and discussion parts of the paper to better bring these ideas to light.
>
> [1] ParaRNN: https://arxiv.org/pdf/2510.21450
>
> [2] https://arxiv.org/abs/2404.08819
>
> [3] https://arxiv.org/abs/2411.12537

---

> > ### Author Rebuttal · Reviewer_zax9 · 2026-04-01
> >
> > Thanks for your response. This is a paper in good shape, and I will maintain my current score.

---

### Official Review · Reviewer_utXz · 2026-03-11

**Soundness:** 3
**Presentation:** 3
**Significance:** 3
**Originality:** 3
**Overall Recommendation:** 4
**Confidence:** 2

**Summary:**

This paper establishes a tight connection between types of RNNs and standard complexity class ($NC^1$, $PNC^1$, $L$, $P$), thereby theoretically explaining why linear RNN (LRNN) is more amenable to parallelize in practice compared to traditional nonlinear RNNs. It further points out fine-grained expressivity differences between two types of LRNNs, i.e., DPLR and PD. The experiments verified the theoretical conclusions, providing a foundation for designing LLM architectures that achieve an optimal balance between expressivity and parallelism.

**Compliance With Llm Reviewing Policy:**

Affirmed.

**Final Justification:**

The rebuttal did not substantially change my overall evaluation, but it did not raise additional concerns either.  I therefore maintain my current score of Weak Accept.

**Key Questions For Authors:**

1. In Theorem 5, I don't understand why it is 4-layer. Is this bound related to the construction of the number of layers?

**Limitations:**

Yes.

**Strengths And Weaknesses:**

Strengths

1. From the perspective of complexity, it reveals the fundamental trade-off in expressivity and parallelism between linear and nonlinear RNNs.

2. The fine-grained expression capabilities of different LRNN variants in $PNC^1$ were compared, which is more informative than the coarse-grained comparison of the strengths/weaknesses of Transformers.

3. The experiment has confirmed that their results about RNNs’ expressivity predict their behavior when trained on synthetic tasks.

Weaknesses

1. There are some spelling mistakes in the text. For example: "differents" on line 334, "succintly" on line 348, etc.

---

> ### Author Rebuttal · Authors · 2026-03-30
>
> Thanks for your positive review!
>
> We will address the spelling mistakes you identified.
>
> For Theorem 5, if the transition matrices can have an arbitrary form, the construction would be possible with a single layer. However, since the transition matrices have DPLR form, we need a few extra layers to factor blocks of transition matrices into blocks of DPLR matrices, following a neat idea from Peng et al. (2025). The full proof in Appendix C.3 details how this can be achieved in 4 total layers, 3 of which do the factorization, and one of which multiplies all the factored transitions.
>
> We hope this addresses your concerns, but we are happy to address any further questions.

---

> > ### Author Rebuttal · Reviewer_utXz · 2026-04-01
> >
> > Thank you for the clarification.  It would be helpful to state this more explicitly in Theorem 5 for readability.  I will maintain my current score.

---

### Official Review · Reviewer_mByq · 2026-03-12

**Soundness:** 4
**Presentation:** 3
**Significance:** 3
**Originality:** 3
**Overall Recommendation:** 5
**Confidence:** 3

**Summary:**

The paper analyzes why linear RNNs (LRRNs) are more parallelizeable than the standard, non-linear RNNs. The authors show through a circuit complexity argument that LRNNs are almost as parallelizeable as transformers, incurring only a small depth overhead when encoded as a circuit. In contrast, nonlinear RNNs can compute L-complete or even P-complete problems, which means that there is a fundamental barrier when parallelizing them. The authors also show that different variants of LRNNs have different expressive power: diagonal-plus-low-rank LRNNs are more expressive than permutation-diagonal ones. Finally, experiments on graph connectivity tasks and iterated multiplication validate the theoretical findings.

**Compliance With Llm Reviewing Policy:**

Affirmed.

**Final Justification:**

My score was already high and the authors addressed my questions and concerns, therefore I will keep my score.

**Key Questions For Authors:**

1. Do you have any high-level intuitions for theorems 3 and 4?
2. It's not very clear to me what makes the results so different between iterated matrix multiplication over Z_m vs. Z. Could you clarify this?

**Limitations:**

Yes.

**Strengths And Weaknesses:**

Strengths:
1. The paper tackles an interesting and practically relevant research question, which makes it a valuable contribution to the field.

2. The paper is generally clear and well-written, despite being very technical. The authors do a good job at presenting the technical background required to understand the paper. Despite not verifying the proofs in the appendix, it looks like all theoretical results are proven rigorously. In addition, the authors provide intuitive high-level proof sketches in the main text to aid understanding the general idea of the proofs.

3. The theoretical results are accompanied by a suitable set of experiments that validate the theoretical insights.

Weaknesses:

In the current form, I don't see many weaknesses. I spotted the following minor typos: and and (line 107, left col), missing full stop (l85, l99, right col). This is only a suggestion but I encourage the authors to use the extra page to add some more high-level explanations and reiterate the connection between the different complexity classes throughout the paper if it is accepted. Also, I think the motivation would be stronger if you expanded a bit more on why LRNNs are practically relevant. Currently there is only a sentence mentioning this in the introduction. Finally, I think the introduction would be more accessible if you defined intuitively the different complexity classes somehow. Figure 1 does a great job at showing how the different results relate to each other, but until then, the summary is a bit confusing.

---

> ### Author Rebuttal · Authors · 2026-03-30
>
> Thanks for your review! We will fix the typos you identified.
>
> We agree that adding more high-level explanation of complexity classes would improve exposition and will make efforts to incorporate this in revisions. Regarding practical relevance, linear RNNs and hybrid models have recently gained wide practical adoption, though precise details of the underlying architecture are still somewhat open. For instance, whereas some large-scale releases (Qwen 3.5, Kimi Linear, Olmo Hybrid) use linear RNNs like GatedDeltaNet, others recent architecture work (e.g., xLSTM and M2RNN) re-incorporates nonlinear RNN components or methods to parallelize nonlinear RNNs [1]. Our results help reveal the fundamental tradeoff between these approaches, and we hope they could help guide the convergence of these directions in the long term, similar to how past theoretical work on linear RNN expressiveness [2, 3] helped inspire innovations in the transition matrices for modern “slightly non-diagonal” linear RNNs. We will revise the paper to better highlight this motivation and potential connection to practice.
>
> [1] ParaRNN: https://arxiv.org/pdf/2510.21450
>
> [2] https://arxiv.org/abs/2404.08819
>
> [3] https://arxiv.org/abs/2411.12537

---

> > ### Author Rebuttal · Reviewer_mByq · 2026-04-01
> >
> > I don't have any further concerns. I will maintain my current score.

---

### Official Review · Reviewer_j2Xy · 2026-03-15

**Soundness:** 4
**Presentation:** 4
**Significance:** 3
**Originality:** 4
**Overall Recommendation:** 5
**Confidence:** 5

**Summary:**

This paper presents a theoretical study over the representation power for different RNN architecture including permutation-diagonal LRNN, diagonal-plus-low-rank LRNNs, and non-linear RNNs and separate (the complexity of running) them into three classes in complexity theory.

1. permutation-diagonal LRNN is NC$^1$ complete while falls in NC$^1$.

2. diagonal-plus-low-rank LRNNs is PNC$^1$ complete while falls in PNC$^1$, therefore is potentially beyond NC$^1$.

3. general non-linear RNNs is L-complete is log precision and P-complete in poly precision.

The theoretical result also predicts the performance of different architecture on synthetic tasks.

**Compliance With Llm Reviewing Policy:**

Affirmed.

**Key Questions For Authors:**

Please refer to strength and weakness for my questions. Also some minor comments include

1. This paper seems to miss some earlier work in the representation power of RNNs including [1,2,3]. [1] discusses a closely related result on why bounded-precision RNNs are Turing complete. [2,3] dicusses the implication of bounded memory on the complexity class of RNNs, which are conceptually related to Proposition 1.

[1] Turing Completeness of Bounded-Precision Recurrent Neural Networks

[2] RNNs are not Transformers (Yet): The Key Bottleneck on In-context Retrieval

[3] Representational Strengths and Limitations of Transformers

**Limitations:**

yes

**Strengths And Weaknesses:**

**Soundness**. The theoretical results are very convincing. The synthetic experiments incorporate the theory well. The reviewer only have three minor questions:

1. Why permutation-diagonal LRNN is omitted in the experimental study?

2. The authors cite previous works proving that Transformers is in TC$^0$, why the authors consider the computation depth under NC circuit depth instead of TC circuit depth?

3. The comparison between circuit depth and actual model depth near line 212 is confusing as (1) this seems to be two different notion, one is the complexity to simulate one layer and the other is the number of layer; and (2) the final result on 256 to 400 layers is not unimaginable especially given advancement in techniques to stably train deep networks.

**Presentation**. This paper is very well-written and clear.

**Significance**. This paper makes a solid technical contribution in precisely characterization the complexity of different linear RNN architecture.

**Originality**. The result is very surprising as it is not conjectured that it is possible to separate permutation-diagonal and diagonal + low rank LRNNs.

---

> ### Author Rebuttal · Authors · 2026-03-30
>
> Thanks for your review!
>
> We did not do experiments with PD LRNNs because they are not implemented in the [Flash Linear Attention](https://github.com/fla-org/flash-linear-attention) repository that we used to replicate each architecture. We agree that would be interesting to run experiments with PD LRNNs but found it somewhat painful in practice, and therefore we chose to prioritize other parts of the paper.
>
> We normalize circuit depth in terms of NC circuits because that is closest to actual hardware. Hardware circuits consist of binary logic gates like AND or OR like those found in NC circuits. In contrast, TC0 allows unbounded arity MAJ gates, which are not supported on hardware. Thus, while TC0 is a robust theoretical class useful in expressivity analyses, it would incur a log factor depth overhead to implement a TC0 circuit using more practical NC1-style circuits (in this sense, using an architecture in TC0 is overly constraining). Viewing models via the practical lens of NC-style circuits thus allows us to make the important observation that LRNNs, although capable of expressing log-depth computation in NC1 and even PNC1, are in fact no less parallelizable than transformers.
>
> Regarding point 3, our analysis indeed describes the circuit depth required to simulate one (or a fixed number) of layers. Taking the number of layers in the model as fixed with respect to context length n (as is the case when applying an already trained model to adaptively longer inputs) it thus describes the circuit depth required to simulate a model of fixed depth on an input of length n. We will update the discussion around line 212 to make this clear.

---

> > ### Author Rebuttal · Reviewer_j2Xy · 2026-04-01
> >
> > I agree with the authors on the finite fan in arguments. Some of additional experiments on PD LRNN would be helpful but it is not crucial for this paper

---

### Decision · Program_Chairs · 2026-04-30

**Decision:**

Accept (regular)

**Comment:**

I don’t particularly appreciate expressivity papers anymore; however, this is clearly a very good, thorough, and non-trivial paper.